# How Different from the Past?
# Spatio-Temporal Time Series Forecasting with
# Self-Supervised Deviation Learning

**Haotian Gao**[1]*, **Zheng Dong**[1]*, **Jiawei Yong**[2]
**Shintaro Fukushima**[2], **Kenjiro Taura**[1], **Renhe Jiang**[1]†
[1]The University of Tokyo, [2]Toyota Motor Corporation
{gaoht6, zhengdong00}@outlook.com
{jiawei_yong, s_fukushima}@mail.toyota.co.jp
tau@eidos.ic.i.u-tokyo.ac.jp, jiangrh@csis.u-tokyo.ac.jp

## Abstract

Spatio-temporal forecasting is essential for real-world applications such as traffic management and urban computing. Although recent methods have shown improved accuracy, they often fail to account for dynamic deviations between current inputs and historical patterns. These deviations contain critical signals that can significantly affect model performance. To fill this gap, we propose **ST-SSDL**, a Spatio-Temporal time series forecasting framework that incorporates a Self-Supervised Deviation Learning scheme to capture and utilize such deviations. ST-SSDL anchors each input to its historical average and discretizes the latent space using learnable prototypes that represent typical spatio-temporal patterns. Two auxiliary objectives are proposed to refine this structure: a contrastive loss that enhances inter-prototype discriminability and a deviation loss that regularizes the distance consistency between input representations and corresponding prototypes to quantify deviation. Optimized jointly with the forecasting objective, these components guide the model to organize its hidden space and improve generalization across diverse input conditions. Experiments on six benchmark datasets show that ST-SSDL consistently outperforms state-of-the-art baselines across multiple metrics. Visualizations further demonstrate its ability to adaptively respond to varying levels of deviation in complex spatio-temporal scenarios. Our code and datasets are available at https://github.com/Jimmy-7664/ST-SSDL.

## 1   Introduction

Accurately forecasting the spatio-temporal time series generated from various sensors is essential for a wide range of applications like traffic flow regulation, energy demand estimation, and climate impact analysis. Due to the complex dependencies across time and space, considerable efforts have been made to learn rich spatio-temporal patterns [7, 14, 44, 49, 63, 77, 78, 83, 24, 21, 58, 56, 57, 19, 20, 52, 34, 16, 74, 75, 11, 15, 12].

However, a crucial aspect still remains overlooked by current models: *the deviation between current observations and their historical states.* In real-world transportation systems, the present time series frequently differ from historical states due to policy interventions, special events, or external incidents. These deviations can provide valuable insights for forecasting, as they often signal changes that impact future behaviors. Although some recent efforts [40, 42, 24, 54] attempt to incorporate

---

*Equal contribution.
†Corresponding author.

39th Conference on Neural Information Processing Systems (NeurIPS 2025).

historical context by using fixed temporal offsets, such as observations from the same time on previous days or weeks, these methods struggle to capture the dynamic deviations, limiting their ability to respond effectively when current patterns diverge from history. As shown in Figure 1(a), the degree of deviation dynamically varies with spatio-temporal contexts. For example, a traffic sensor in downtown often shows high variance, as indicated by the green lines, while a rural road can remain relatively stable, as shown by the blue pairs. Therefore, modeling dynamic deviations remains a challenge in spatio-temporal forecasting. A straightforward approach to do this is to compute the distance between the current input and its historical average, flagging deviations when this distance exceeds a threshold [69, 79]. However, this strategy treats deviation as a binary event, whereas in reality, it evolves continuously. It is difficult to precisely quantify how the current input differs from the past, especially when encoded into high-dimensional latent space. As visualized in Figure 1(b), while the deviations of the green pairs $D_1$ and $D_2$ appear evident in the physical space (i.e., $D_1 \approx 40$, $D_2 \approx 20$), it is uncertain how much the corresponding distance $\widetilde{D}_1$ and $\widetilde{D}_2$ should be in the latent space. These observations highlight our key challenge: **how to quantify dynamic spatio-temporal deviations in continuous latent space and leverage them to enhance spatio-temporal forecasting.**

To address the above challenge, we introduce our core idea: **relative distance consistency**. Intuitively, the current-history pairs that are close (far) in the physical space should remain close (far) in latent space, i.e., $D_1 > D_2 \Rightarrow \widetilde{D}_1 > \widetilde{D}_2$. Building upon this idea, we propose Self-Supervised Deviation Learning (**SSDL**) that enables the spatio-temporal model to capture spatio-temporal deviations in a fully self-supervised manner without any prior knowledge. Specifically, we first utilize the historical average of each current input as a self-supervised anchor. Then, we discretize the continuous latent space by using a set of learnable prototypes, where a contrastive loss is employed to guide the learning and discretization process. Next, we map each input and its anchor to their

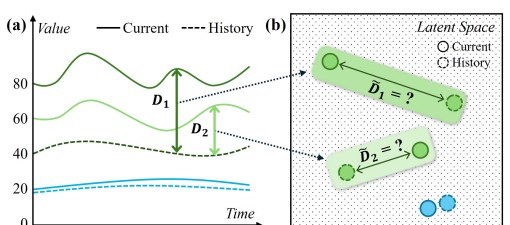

Figure 1: (a) Deviations between current and historical states vary with spatio-temporal context. (b) Such deviations in latent space are hard to quantify, so we leverage **relative distance consistency**: current–history pairs that are close (far) in physical space should remain close (far) in latent space, i.e., $D_1 > D_2 \Rightarrow \widetilde{D}_1 > \widetilde{D}_2$.

nearest prototypes via cross-attention, treating each prototype as the proxy of the latent representation. The distance between the proxy prototypes of the current input and its historical anchor then serves as an approximation of the latent-space deviation. Finally, we propose a self-supervised deviation loss to minimize the discrepancy between the physical-space distance $D$ and its latent-space counterpart $\widetilde{D}$, thereby enforcing the principle of relative distance consistency, i.e., $D \uparrow \Rightarrow \widetilde{D} \uparrow$, $D \downarrow \Rightarrow \widetilde{D} \downarrow$. Our contributions are summarized as follows:

- We propose **SSDL**, the first self-supervised method designed to learn deviations in spatio-temporal time series. It is implemented with two self-supervised objectives: a contrastive loss to discretize the continuous latent space and a deviation loss to enforce relative distance consistency between the physical and latent spaces.

- We propose **ST-SSDL**, a novel spatio-temporal forecasting framework that incorporates self-supervised deviation learning to enhance spatio-temporal forecasting performance, improving adaptability to various deviation levels.

- We validate ST-SSDL through experiments on six benchmark datasets, where it achieves state-of-the-art performance. Comprehensive ablation studies and visualized case analyses further confirm that the model adapts its latent space based on deviation levels.

## 2 Related Work

### 2.1 Spatio-Temporal Forecasting

Traditional statistical methods [10, 26, 55, 68] primarily focus on capturing temporal dependencies. With the rise of deep learning, numerous spatio-temporal forecasting models have been proposed to better capture spatio-temporal relations. Early work emphasizes graph neural networks (GNNs), which naturally align with sensor network structures [44, 77, 83, 24, 67, 2, 65, 35, 17, 3, 18].

Recently, inspired by the success of Transformer [71], a line of attention-based models [33, 49, 36, 38, 81, 76] has shown strong performance in spatio-temporal forecasting. To address the quadratic time complexity of attention mechanism, Mamba [23] introduces a linear-time sequence modeling framework, which has drawn a series of Mamba-based models [29, 66, 70, 43, 39] for spatio-temporal tasks. Beyond these complex designs, several studies [63, 73, 72, 86] have explored the lightweight MLP-based models in modeling spatio-temporal data. *However, these existing approaches overlook the deviations between current and historical patterns under dynamic spatio-temporal conditions.*

## 2.2 Self-Supervised Learning

Self-supervised learning (SSL) has become a powerful paradigm across vision, language, and beyond. In computer vision, contrastive methods [6, 27, 22], alongside masked modeling approaches [28, 80], achieve strong performance without labels. In NLP, self-supervised pretraining strategies [37, 51, 9] also have demonstrated their effectiveness in language modeling tasks. These advances also extend to speech [30, 1, 45], multimodal [82, 25], and time series domains [50, 61]. Recent studies have explored the potential of self-supervised learning in spatio-temporal data. Specifically, several methods based on self-supervised masked modeling have shown promising results [64, 21, 48, 47, 32, 84]. In addition, self-supervised contrastive learning frameworks [85, 13, 59, 31] have also been developed, demonstrating their effectiveness in learning robust spatio-temporal representations. *However, these methods fail to explicitly model the dynamic deviation in spatio-temporal data. Our method fills this gap through a self-supervised deviation learning approach that enhances robustness without relying on additional supervision.*

## 3 Problem Definition

Given $N$ spatially distributed sensors, at every timestep $t$, each sensor has a $C$-dimensional observation $X \in \mathbb{R}^C$. For an input tensor $X^c \in \mathbb{R}^{T \times N \times C} = X_{t-T+1:t}$ that contains the most recent $T$ steps, the forecasting task is to predict the next $T'$ steps for all nodes and channels:

$$\widehat{X}_{t+1:t+T'} = F_\theta\big(X_{t-T+1:t}\big), \qquad F_\theta : \mathbb{R}^{T \times N \times C} \to \mathbb{R}^{T' \times N \times C} \tag{1}$$

where $F_\theta$ is a learnable spatio-temporal model parameterized by $\theta$. In our study, C is equal to 1 in the used benchmarks for spatio-temporal time series forecasting [44, 67, 83].

## 4 Methodology

### 4.1 Self-Supervised Deviation Learning

Real-world spatio-temporal data often exhibit informative deviations from historical patterns which are critical for forecasting but typically overlooked by existing methods. Moreover, such deviations are not categorical events but emerge as continuous, context-dependent variations across time and space, making them difficult to capture using binary method like threshold. To tackle these challenges of modeling spatio-temporal deviations, we propose Self-Supervised Deviation Learning (**SSDL**). It anchors current inputs to historical averages and discretizes their latent differences via learnable prototypes. Two auxiliary self-supervised objectives further guide the model to organize latent space and quantify deviations, improving adaptability across diverse input conditions.

**History as Self-Supervised Anchor.** A core challenge in modeling deviation lies in the absence of a principled reference: without a contextual baseline, dynamic deviations become hard to define and quantify. To address this, we introduce historical averages as anchors as illustrated in Figure 2(a). These anchors summarize recurring spatio-temporal patterns and serve as references for deviation modeling. Concretely, the full training sequence $X^{\text{train}} \in \mathbb{R}^{T^{\text{all}} \times N \times C}$ is partitioned into $S$ non-overlapping weekly segments based on the periodicity of spatio-temporal data. They each contain $T^w$ timesteps, resulting in a tensor $X^w \in \mathbb{R}^{S \times T^w \times N \times C}$. The historical anchor $\bar{X}^w \in \mathbb{R}^{T^w \times N \times C}$ is then computed by averaging aligned timesteps across all weeks as $\bar{X}^w = \frac{1}{S} \sum_{s=1}^{S} X_s^w$. For the current input $X^c \in \mathbb{R}^{T \times N \times C}$, we retrieve its timestamp-aligned historical anchor $X^a \in \mathbb{R}^{T \times N \times C}$ from $\bar{X}^w$. Both sequences are then encoded via a shared function $f^{\text{enc}}(\cdot)$, producing latent representations $H^c$

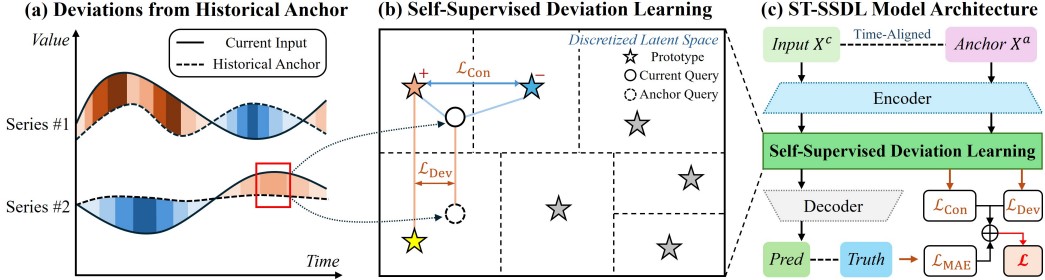

Figure 2: The overview of our proposed framework **ST-SSDL**: **S**patio-**T**emporal Time Series Forecasting with **S**elf-**S**upervised **D**eviation **L**earning.

and $H^a \in \mathbb{R}^{N \times h}$ with hidden dimension $h$:

$$
\begin{aligned}
H^c &= f^{\text{enc}}(X^c) \\
H^a &= f^{\text{enc}}(X^a)
\end{aligned}
\tag{2}
$$

This anchoring process equips the model with a context-aware pivot, offering a structured reference against which deviations can be systematically quantified. By serving as intrinsic supervision signals, these anchors guide the model to learn deviation-aware representations without external labels. As a result, grounding hidden representations in historical context supports the formation of a structured latent space, forming a concrete basis for self-supervised modeling of dynamic deviations.

**Self-Supervised Space Discretization.** Deviations in spatio-temporal data exhibit diverse and complex dynamics, making them difficult to measure directly within the continuous latent space. To mitigate this, we introduce a set of $M$ learnable prototypes $\mathbf{P}_1, \ldots, \mathbf{P}_M \in \mathbb{R}^{M \times d}$ with hidden dimension $d$ that serve as representative spatio-temporal patterns as shown in Figure 2(b). While prior works [5, 46, 53] mainly focus on learning such prototypes to capture typical behaviors, we additionally leverage them to discretize the latent space. This enables structured comparison between current and historical representations. **The latent space discretized by prototypes** provides a foundation for quantifying deviation. Moreover, we establish interaction between inputs and the prototypes via a query-prototype attention mechanism. For a query $Q \in \mathbb{R}^d$ projected from input hidden representation, the attention scores over the prototypes are computed as follows:

$$
\boldsymbol{\alpha}_i = \frac{\exp\left(\left(Q \cdot \mathbf{P}_i^\top\right)/\sqrt{d}\right)}{\sum_{j=1}^M \exp\left(\left(Q \cdot \mathbf{P}_j^\top\right)/\sqrt{d}\right)}
\tag{3}
$$

where $\boldsymbol{\alpha}_i$ denotes the attention weight assigned to the $i$-th prototype $\mathbf{P}_i$. Higher attention scores indicate stronger affinity between the query and the corresponding prototype, reflecting the underlying similarity in spatio-temporal characteristics.

Based on the computed attention scores, we sort the prototypes in descending order for each query. For current input hidden representation $H^c$, the query $Q^c$ is obtained by applying a linear projection to calculate the attention scores. The resulting ranking reveals how the input matches with the prototype structure. This prototype-based discretization transforms unstructured deviations into interpretable patterns, facilitating accurate measurement and comparison under diverse spatio-temporal dynamics.

To guide the prototype organization, we define a contrastive objective that promotes discriminability among prototypes. Specifically, we select the most relevant and second-most relevant prototypes as the positive sample $\mathcal{P}^c$ and negative sample $\mathcal{N}^c$. These selections serve as supervision targets in the contrastive learning by defining which prototypes should be drawn closer or pushed apart. A variant of the triplet loss [60] is adopted as:

$$
\mathcal{L}_{\text{Con}} = \max\left(\|\widetilde{\nabla}(Q^c) - \mathcal{P}^c\|_2^2 - \|\widetilde{\nabla}(Q^c) - \mathcal{N}^c\|_2^2 + \delta, 0\right)
\tag{4}
$$

where $\delta$ is a positive margin and $\widetilde{\nabla}$ denotes the stop-gradient operation applied to the query. This contrastive formulation encourages each prototype to pull closer its most semantically aligned query while pushing away less relevant one, resulting in inter-prototype separability. Moreover, this

mechanism partitions the latent space into multiple prototype-centered regions, which can be viewed like discrete bins as illustrated in Figure 2(b). As a consequence, the latent space becomes structurally organized, with distinct clusters of queries forming around their associated prototypes. Such design enhances the model's ability to distinguish inputs based on their alignment with prototypes.

**Self-Supervised Deviation Quantification.** With the prototype-based discretization offering a structured view of spatio-temporal patterns, we can design an explicit training signal to enforce **relative distance consistency**. To this end, we propose a self-supervised learning strategy that leverages prototype rankings from both current representation $H^c$ and historical representation $H^a$ to guide latent organization via a deviation loss.

The deviation loss promotes relational consistency between input representations and prototypes in a distance-of-distances manner. Specifically, for the historical input, the query $Q^a$ is obtained by linearly projecting its hidden representation $H^a$. The corresponding positive prototype $\mathcal{P}^a$ is then selected using the same ranking procedure as applied to $Q^c$. Lastly, we compute the $L_1$ distance between $Q^c$ and $Q^a$, and enforce the distance between their corresponding positive prototypes $\mathcal{P}^c$ and $\mathcal{P}^a$ to match it:

$$\mathcal{L}_{\mathrm{Dev}} = \left\| \widetilde{\nabla}(\|Q^c - Q^a\|_1) - \|\mathcal{P}^c - \mathcal{P}^a\|_1 \right\|_1 \tag{5}$$

By stopping the gradient, $\widetilde{\nabla}(\|Q^c - Q^a\|_1)$ can be taken as an approximate of the physical-space distance $D$ for the current input $X^c$ and its historical anchor $X^a$. With the nearest prototypes as proxies, $\|\mathcal{P}^c - \mathcal{P}^a\|_1$ can represent their latent-space distance $\widetilde{D}$. As a result, $\mathcal{L}_{\mathrm{Dev}}$ ensures relative distance consistency by minimizing $\|D - \widetilde{D}\|_1$, thus inputs with similar semantics are routed to the same prototype or neighboring prototypes, while those with distinct patterns are mapped further apart. Meanwhile, the $\widetilde{\nabla}$ operator in $\mathcal{L}_{\mathrm{Con}}$ and $\mathcal{L}_{\mathrm{Dev}}$ also prevents the model from entering *lazy* mode where all representations become overly similar and map to the same prototype.

Together, these two objectives refine the prototypes for deviation-aware learning. The contrastive loss shapes inter-prototype separation, while the deviation loss promotes deviation consistency between input representations and corresponding prototypes in latent space. This joint formulation enables the model to quantify dynamic deviations in a fully self-supervised manner.

## 4.2 Enhancing Spatio-Temporal Forecasting with Self-Supervised Deviation Learning

Current spatio-temporal forecasting models often rely solely on mapping past observations to future targets, overlooking potential deviations between the current input and its historical context. Such dynamic deviations offer significant information that can improve predictive accuracy. As demonstrated in Figure 2(c), building upon the SSDL method introduced above, we propose **ST-SSDL**, a spatio-temporal forecasting framework that incorporates it as a core component.

Recent advances in spatio-temporal modeling [44, 2, 41, 62, 35, 17] have demonstrated the effectiveness of injecting graph convolutional operations into recurrent neural architectures. These designs culminate in Graph Convolution Recurrent Units (GCRUs), which jointly capture spatial dependencies encoded by the graph topology and temporal dynamics intrinsic to time series. Therefore, ST-SSDL follows this paradigm through a GCRU-based architecture to effectively capture spatio-temporal dependencies. In detail, we use GCRU as the encoding function $f^{\mathrm{enc}}$ in Equation (2). The details of GCRU are formally defined as follows:

$$\begin{cases} r_t = \sigma\left([X_t \,|\, H_{t-1}] \star_{\mathcal{G}} \Theta_r + b_r\right) \\ u_t = \sigma\left([X_t \,|\, H_{t-1}] \star_{\mathcal{G}} \Theta_u + b_u\right) \\ c_t = \tanh\left([X_t \,|\, r_t \odot H_{t-1}] \star_{\mathcal{G}} \Theta_c + b_c\right) \\ H_t = u_t \odot H_{t-1} + (1 - u_t) \odot c_t \end{cases} \tag{6}$$

where $X_t \in \mathbb{R}^N$ and $H_t \in \mathbb{R}^{N \times h}$ represents input and output at timestep $t$. $r_t$, $u_t$, and $c_t$ are the reset gate, update gate, and candidate state, respectively. $\odot$ and $|$ denote element-wise multiplication and feature concatenation. The operator $\star_{\mathcal{G}}$ applies graph convolution over graph $\mathcal{G}$ defined as:

$$Z \star_{\mathcal{G}} \Theta = \sum_{k=0}^{K} \tilde{A}^k Z W_k \tag{7}$$

where $Z \in \mathbb{R}^{N \times h}$ is the input of graph convolution operation. $\tilde{\mathcal{A}} \in \mathbb{R}^{N \times N}$ denotes the topology of graph $\mathcal{G}$ and $W_k \in \mathbb{R}^{K \times h \times h}$ is Chebyshev polynomial weights to order $K$.

The proposed model follows an encoder-decoder architecture built by stacking multiple GCRU cells. Each input consists of a current observation sequence $X^c$ and its aligned historical anchor $X^a$, both enriched with input embeddings, node embeddings and time-of-day embeddings [77, 63, 49] to better capture node-specific semantics and periodic temporal patterns. As described in Equation (2), both sequences are processed in parallel through the encoder. Specifically, the input is passed through the GCRU encoder to get the hidden state at the last timestep $H^c = H_t$. Similarly, we can get $H^a$ from $X^a$. Each of these representations is then refined through query-prototype attention in Equation (3), where the enhanced representation is computed as a weighted sum of all prototypes: $V = \sum_{i=1}^{M} \boldsymbol{\alpha}_i \mathbf{P}_i$. This process yields augmented representations $V^c$ and $V^a$ for $H^c$ and $H^a$, respectively. We then concatenate these four components and pass them through a linear projection to generate the adaptive adjacency matrix $\tilde{\mathcal{A}}$, enabling the decoder to adaptively model spatial interactions conditioned on the encoded spatio-temporal context. This process can be formulated as:

$$
\begin{aligned}
H' &= W \left[ H^c \,|\, V^c \,|\, H^a \,|\, V^a \right] + b \\
\tilde{\mathcal{A}} &= \mathrm{Softmax}(\mathrm{ReLU}(H' \cdot H'^{\top}))
\end{aligned}
\tag{8}
$$

where $W$ and $b$ are learnable parameters of linear projection. Finally, a GCRU decoder takes the adaptive graph and the hidden states generated by encoder to make the future prediction sequence.

The primary objective of ST-SSDL is to achieve accurate spatio-temporal forecasting, supervised by the Mean Absolute Error (MAE) between predictions and ground truth values. To enhance deviation-aware modeling, the model integrates the contrastive loss $\mathcal{L}_{\mathrm{Con}}$ and deviation loss $\mathcal{L}_{\mathrm{Dev}}$ introduced in Section 4.1, together with the MAE loss. Finally, the overall training objective is defined as:

$$
\mathcal{L} = \mathcal{L}_{\mathrm{MAE}} + \lambda_{\mathrm{Con}} \cdot \mathcal{L}_{\mathrm{Con}} + \lambda_{\mathrm{Dev}} \cdot \mathcal{L}_{\mathrm{Dev}}
\tag{9}
$$

where $\lambda_{\mathrm{Con}}$ and $\lambda_{\mathrm{Dev}}$ are hyper-parameters that control the contributions of the contrastive and deviation terms. This joint loss encourages accurate forecasting while adaptively capturing spatio-temporal deviations to improve robustness across diverse scenarios.

**Complexity Analysis.** We analyze the complexity of ST-SSDL by examining its two main components: the SSDL method and the GCRU backbone. In SSDL, query-prototype attention over $M$ prototypes yields a complexity of $\mathcal{O}(NMd)$. For the GCRU backbone, each layer applies a $K$-order Chebyshev graph convolution and recurrent update, resulting in $\mathcal{O}(KNh^2)$ per timestep. With $L$ GCRU layers and total sequence length $T + T'$, the overall cost of GCRU backbone is $\mathcal{O}(LKNh^2(T + T'))$. Thus, the total complexity of ST-SSDL is $\mathcal{O}(NMd + LKNh^2(T + T'))$.

# 5 Experiment

## 5.1 Experimental Setup

**Datasets.** We evaluate our ST-SSDL on 6 widely used spatio-temporal forecasting benchmarks: METRLA, PEMSBAY, PEMSD7(M) [44, 83] with traffic speed data, and PEMS04, PEMS07, PEMS08 [67] with traffic flow data. All datasets have a temporal resolution of 5 minutes per timestep (12 steps per hour). More statistics for these benchmarks are provided in Table 1. Following prior work, METRLA and PEMSBAY adopt a 70%:10%:20% split [77] for training, validation, and testing, respectively, while PEMSD7(M), PEMS04, PEMS07, and PEMS08 use 60%:20%:20% [24, 49].

**Settings.** We conducted a hyper-parameter search for each benchmark to ensure optimal model performance. The architecture consists of a 1-layer encoder and 1-layer decoder (*i.e.*, $L$=2), with hidden dimensions $h$ set to 128, 64, or 32, depending on the dataset. We use $M$=20 prototypes with dimension $d$=64. Detailed configurations are provided in our public code.

Table 1: Summary of our used spatio-temporal benchmarks.

| Dataset | #Sensors (N) | #Timesteps | Time Range |
|---|---|---|---|
| METRLA | 207 | 34,272 | 03/2012 - 06/2012 |
| PEMSBAY | 325 | 52,116 | 01/2017 - 06/2017 |
| PEMSD7(M) | 228 | 12,672 | 05/2012 - 06/2012 |
| PEMS04 | 307 | 16,992 | 01/2018 - 02/2018 |
| PEMS07 | 883 | 28,224 | 05/2017 - 08/2017 |
| PEMS08 | 170 | 17,856 | 07/2016 - 08/2016 |

The input and prediction horizons are both set to 1 hour ($T$=$T'$=12 timesteps). For optimization, we employ the Adam optimizer with an initial learning rate of 0.001. Model evaluation is based on three key metrics: Mean Absolute Error (MAE), Root Mean

Table 2: Performance on METRLA, PEMSBAY, PEMSD7(M), and PEMS04/07/08 benchmarks.

| Dataset | Metric | HI | GRU | STGCN | DCRNN | GWNet | MTGNN | AGCRN | GTS | STNorm | STID | ST-WA | MegaCRN | STDN | ST-SSDL |
|---|---|---|---|---|---|---|---|---|---|---|---|---|---|---|---|
| METRLA Step 3 15 min | MAE | 6.80 | 3.07 | 2.75 | 2.67 | 2.69 | 2.69 | 2.85 | 2.75 | 2.81 | 2.82 | 2.89 | 2.62 | 2.83 | **2.60** |
| | RMSE | 14.21 | 6.09 | 5.29 | 5.16 | 5.15 | 5.16 | 5.53 | 5.27 | 5.57 | 5.53 | 5.62 | 5.04 | 5.84 | **5.02** |
| | MAPE | 16.72% | 8.14% | 7.10% | 6.86% | 6.99% | 6.89% | 7.63% | 7.12% | 7.40% | 7.75% | 7.66% | 6.68% | 7.78% | **6.62%** |
| METRLA Step 6 30 min | MAE | 6.80 | 3.77 | 3.15 | 3.12 | 3.08 | 3.05 | 3.20 | 3.14 | 3.18 | 3.19 | 3.25 | 3.01 | 3.21 | **2.96** |
| | RMSE | 14.21 | 7.69 | 6.35 | 6.27 | 6.20 | 6.13 | 6.52 | 6.33 | 6.59 | 6.57 | 6.61 | 6.13 | 6.92 | **6.04** |
| | MAPE | 16.72% | 10.71% | 8.62% | 8.42% | 8.47% | 8.16% | 9.00% | 8.62% | 8.47% | 9.39% | 9.22% | 8.18% | 9.40% | **7.92%** |
| METRLA Step 12 60 min | MAE | 6.80 | 4.88 | 3.60 | 3.54 | 3.51 | 3.47 | 3.59 | 3.59 | 3.57 | 3.55 | 3.68 | 3.48 | 3.57 | **3.37** |
| | RMSE | 14.21 | 9.75 | 7.43 | 7.47 | 7.28 | 7.21 | 7.45 | 7.44 | 7.51 | 7.55 | 7.59 | 7.31 | 7.80 | **7.17** |
| | MAPE | 16.71% | 14.91% | 10.35% | 10.32% | 9.96% | 9.70% | 10.47% | 10.25% | 10.24% | 10.95% | 10.78% | 9.98% | 10.98% | **9.48%** |
| PEMSBAY Step 3 15 min | MAE | 3.05 | 1.44 | 1.36 | 1.31 | 1.30 | 1.33 | 1.35 | 1.37 | 1.33 | 1.31 | 1.37 | 1.28 | 1.36 | **1.26** |
| | RMSE | 7.03 | 3.15 | 2.88 | 2.76 | 2.73 | 2.80 | 2.88 | 2.92 | 2.82 | 2.79 | 2.88 | 2.71 | 2.96 | **2.65** |
| | MAPE | 6.85% | 3.01% | 2.86% | 2.73% | 2.71% | 2.81% | 2.91% | 2.85% | 2.76% | 2.78% | 2.86% | 2.67% | 2.91% | **2.60%** |
| PEMSBAY Step 6 30 min | MAE | 3.05 | 1.97 | 1.70 | 1.65 | 1.63 | 1.66 | 1.67 | 1.72 | 1.65 | 1.64 | 1.70 | 1.60 | 1.69 | **1.57** |
| | RMSE | 7.03 | 4.60 | 3.84 | 3.75 | 3.73 | 3.77 | 3.82 | 3.86 | 3.77 | 3.73 | 3.81 | 3.68 | 3.94 | **3.59** |
| | MAPE | 6.84% | 4.45% | 3.79% | 3.71% | 3.73% | 3.75% | 3.81% | 3.88% | 3.73% | 3.73% | 3.81% | 3.59% | 3.81% | **3.49%** |
| PEMSBAY Step 12 60 min | MAE | 3.05 | 2.70 | 2.02 | 1.97 | 1.99 | 1.95 | 1.94 | 2.06 | 1.92 | 1.91 | 2.00 | 1.89 | 1.92 | **1.86** |
| | RMSE | 7.01 | 6.28 | 4.63 | 4.60 | 4.60 | 4.50 | 4.50 | 4.60 | 4.45 | 4.42 | 4.52 | 4.45 | 4.58 | **4.34** |
| | MAPE | 6.83% | 6.74% | 4.72% | 4.68% | 4.71% | 4.62% | 4.55% | 4.88% | 4.46% | 4.55% | 4.63% | 4.48% | 4.54% | **4.35%** |
| PEMSD7(M) Step 3 15 min | MAE | 5.01 | 2.32 | 2.19 | 2.24 | 2.13 | 2.15 | 2.23 | 2.32 | 2.14 | 2.11 | 2.20 | 2.05 | 2.17 | **2.02** |
| | RMSE | 9.58 | 4.43 | 4.14 | 4.31 | 4.03 | 4.10 | 4.17 | 4.27 | 4.08 | 4.00 | 4.17 | 3.88 | 4.17 | **3.83** |
| | MAPE | 12.31% | 5.40% | 5.19% | 5.18% | 5.06% | 5.05% | 5.36% | 5.42% | 5.11% | 5.07% | 5.28% | 4.75% | 5.32% | **4.74%** |
| PEMSD7(M) Step 6 30 min | MAE | 5.02 | 3.26 | 2.76 | 3.09 | 2.74 | 2.78 | 2.87 | 3.16 | 2.73 | 2.68 | 2.77 | 2.67 | 2.76 | **2.57** |
| | RMSE | 9.58 | 6.43 | 5.50 | 6.17 | 5.42 | 5.62 | 5.69 | 5.96 | 5.55 | 5.39 | 5.50 | 5.34 | 5.61 | **5.18** |
| | MAPE | 12.31% | 8.02% | 6.89% | 7.54% | 6.95% | 6.77% | 7.26% | 7.81% | 6.88% | 6.83% | 6.94% | 6.59% | 7.10% | **6.40%** |
| PEMSD7(M) Step 12 60 min | MAE | 5.02 | 4.62 | 3.30 | 4.31 | 3.33 | 3.33 | 3.45 | 4.28 | 3.22 | 3.19 | 3.30 | 3.27 | 3.26 | **3.08** |
| | RMSE | 9.59 | 8.87 | 6.63 | 8.53 | 6.59 | 6.68 | 6.93 | 8.03 | 6.60 | 6.50 | 6.56 | 6.68 | 6.65 | **6.38** |
| | MAPE | 12.32% | 12.22% | 8.54% | 11.10% | 8.74% | 8.35% | 9.02% | 11.42% | 8.36% | 8.48% | 8.40% | 8.50% | 8.65% | **7.96%** |
| PEMS04 Average | MAE | 42.35 | 25.55 | 19.57 | 19.63 | 18.53 | 19.17 | 19.38 | 20.96 | 18.96 | 18.38 | 19.06 | 18.40 | | **18.08** |
| | RMSE | 61.66 | 39.71 | 31.38 | 31.26 | 29.92 | 31.70 | 31.25 | 32.95 | 30.98 | 29.95 | 31.02 | 30.54 | 30.22 | **29.64** |
| | MAPE | 29.92% | 17.35% | 13.44% | 13.59% | 12.89% | 13.37% | 13.40% | 14.66% | 12.69% | **12.04%** | 12.52% | 12.71% | 12.50% | 12.36% |
| PEMS07 Average | MAE | 49.29 | 26.74 | 21.74 | 21.16 | 20.47 | 20.89 | 20.57 | 22.15 | 20.50 | 19.61 | 20.74 | 19.70 | 20.65 | **19.19** |
| | RMSE | 71.34 | 42.78 | 35.27 | 34.14 | 33.47 | 34.06 | 34.40 | 35.10 | 34.66 | 32.79 | 34.05 | 32.72 | 34.77 | **32.51** |
| | MAPE | 22.75% | 11.58% | 9.24% | 9.02% | 8.61% | 9.00% | 8.74% | 9.38% | 8.75% | 8.30% | 8.77% | 8.29% | 10.98% | **8.11%** |
| PEMS08 Average | MAE | 34.66 | 19.36 | 16.08 | 15.22 | 14.40 | 15.18 | 15.32 | 16.49 | 15.41 | 14.21 | 15.41 | 14.83 | 14.25 | **13.86** |
| | RMSE | 50.45 | 31.20 | 25.39 | 24.17 | 23.39 | 24.24 | 24.41 | 26.08 | 24.77 | 23.28 | 24.62 | 23.92 | 24.39 | **23.10** |
| | MAPE | 21.63% | 12.43% | 10.60% | 10.21% | 9.21% | 10.20% | 10.03% | 10.54% | 9.76% | 9.27% | 9.94% | 9.64% | 10.67% | **9.17%** |

Square Error (RMSE), and Mean Absolute Percentage Error (MAPE). All following experiments are conducted on NVIDIA RTX 5000 Ada GPUs.

**Baselines.** We evaluate the performance of our model against a comprehensive set of widely used baselines: statistical method HI [10], univariate time series model GRU [7], and spatio-temporal models including STGCN [83], DCRNN [44], Graph WaveNet [77], AGCRN [2], GTS [62], STNorm [14], STID [63], ST-WA [8], MegaCRN [35], and STDN [4].

## 5.2 Performance Evaluation

The comparison results for spatio-temporal forecasting performance are presented in Table 2. In line with prior research, we report the performance at 3, 6, and 12 timesteps for the METRLA, PEMSBAY, and PEMSD7(M) datasets, and the average performance across all 12 predicted timesteps for the PEMS04, PEMS07, and PEMS08 datasets. As a result, our proposed model, ST-SSDL, consistently achieves state-of-the-art performance across all evaluation metrics and datasets with a simple GCRU backbone. This superior performance underscores the effectiveness of our Self-Supervised Deviation Learning approach, which improves the sensitivity and adaptability to varying levels of deviation in complex spatio-temporal scenarios, leading to robust forecasting.

## 5.3 Ablation Study

In this section, we perform a detailed ablation study to systematically assess the contributions of the critical components of ST-SSDL. To this end, we construct several model variants: (1) w/o $\mathcal{L}_{\text{Con}}$: removes the contrastive loss, which is designed to ensure the distinction of prototypes. (2) w/o $\mathcal{L}_{\text{Dev}}$: removes the deviation loss to regularize the consistency between continuous input space and discretized hidden space. (3) w/o $\mathcal{L}_{\text{Con}}, \mathcal{L}_{\text{Dev}}$: removes both the contrastive and deviation losses. (4) w/o SSDL: removes the whole SSDL strategy, downgrading the model to a simple GCRU without any historical information as input. The results in Table 3 show that both $\mathcal{L}_{\text{Con}}$ and $\mathcal{L}_{\text{Dev}}$ are necessary for SSDL. Furthermore, removing both of them leads to much worse performance, while removing the whole SSDL results in the poorest performance as we expected. Besides, we create a naive version of SSDL by removing latent space discretization with prototypes. Accordingly, $\mathcal{L}_{\text{Con}}$ is removed and

the deviation loss is downgraded to $\mathcal{L}_{\text{Dev}}^{\text{Naive}} = \cos\left(\cos(X^c, X^a), \cos(H^c, H^a)\right)$, where the distances are measured by cosine similarity to eliminate the impact of scale. The results confirm that it is hard to learn deviations in the continuous latent space without discretization. All these validate that ST-SSDL is complete and indivisible to achieve superior spatio-temporal forecasting performance.

Table 3: 12 steps average prediction RMSE of the ablations.

| Model | METRLA | PEMSBAY | PEMSD7(M) | PEMS04 | PEMS07 | PEMS08 |
|---|---|---|---|---|---|---|
| w/o $\mathcal{L}_{\text{Con}}$ | 6.16 | 3.65 | 5.18 | 29.81 | 33.16 | 23.31 |
| w/o $\mathcal{L}_{\text{Dev}}$ | 6.07 | 3.62 | 5.14 | 29.84 | 32.68 | 24.09 |
| w/o $\mathcal{L}_{\text{Con}}$, w/o $\mathcal{L}_{\text{Dev}}$ | 6.06 | 3.67 | 5.16 | 29.91 | 32.81 | 24.13 |
| w/o SSDL | 6.09 | 3.54 | 5.22 | 30.01 | 32.90 | 24.96 |
| Naive SSDL | 6.07 | 3.52 | 5.23 | 29.90 | 33.59 | 23.62 |
| **ST-SSDL** | **6.02** | **3.51** | **5.11** | **29.64** | **32.51** | **23.10** |

## 5.4 Efficiency Study

We compare the computational efficiency of ST-SSDL with 5 typical baseline models representing different architectures, including GCN, GCRU, and Transformer-based designs. Table 4 reports the number of parameters, training time (per epoch), and inference time on the METRLA, PEMS07, and PEMS08 datasets. For consistency and fairness, we set batch size $B=16$ to normalize the results. As shown in the table, ST-SSDL has the fewest number of parameters. For example, on PEMS08, it has 100K parameters, which is about 66% of the second-lightest model AGCRN (150K), and only 1.7% of STDN (5876K). However, its runtime efficiency is constrained by the iterative GCRU backbone, which shows a trade-off.

Table 4: Efficiency comparison on METRLA, PEMS07, and PEMS08 datasets.

| Model | METRLA ($N = 207$) | | | PEMS07 ($N = 883$) | | | PEMS08 ($N = 170$) | | |
|---|---|---|---|---|---|---|---|---|---|
| | #Params | Train | Infer | #Params | Train | Infer | #Params | Train | Infer |
| MTGNN | 405K | 62.95s | 1.24s | 1368K | 212.79s | 5.13s | 353K | 28.48s | 0.51s |
| AGCRN | 752K | 87.15s | 2.68s | 755K | 342.73s | 9.89s | 150K | 38.44s | 1.18s |
| STNorm | 224K | 64.80s | 0.88s | 570K | 279.37s | 4.68s | 205K | 28.06s | 0.58s |
| ST-WA | 375K | 113.90s | 7.41s | 786K | 678.93s | 60.99s | 353K | 47.88s | 3.80s |
| STDN | 5971K | 459.43s | 10.39s | 4480K | 693.32s | 32.56s | 5876K | 200.60s | 5.71s |
| **ST-SSDL** | 498K | 71.84s | 8.65s | 379K | 870.12s | 217.80s | 100K | 65.81s | 7.97s |

## 5.5 Hyperparameter Study

In this section, we analyze the impacts of two critical hyper-parameters: the number of prototypes $M$ and their dimension $d$. Figure 3 shows the average RMSE of predictions on METRLA and PEMSBAY datasets when varying $M$ from 5 to 25 and $d$ from 16 to 80. The figure reveals that our model remains fairly stable across these hyper-parameter settings. Using 20 prototypes with a dimension of 64 generally gives good results. However, a very small $M$ or $d$ is insufficient to discretize the deviations in spatio-temporal patterns, while large values can introduce overfitting.

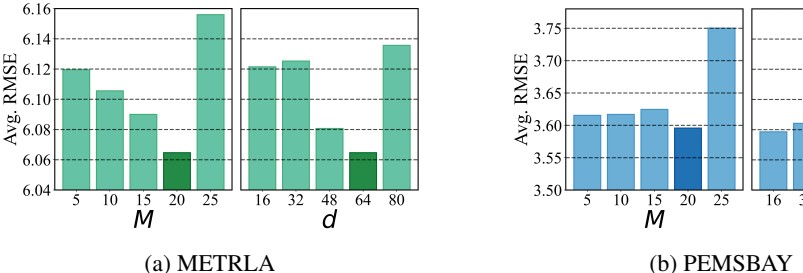

(a) METRLA          (b) PEMSBAY

Figure 3: 12 steps average RMSE w.r.t. #prototypes $M$ and dimension $d$.

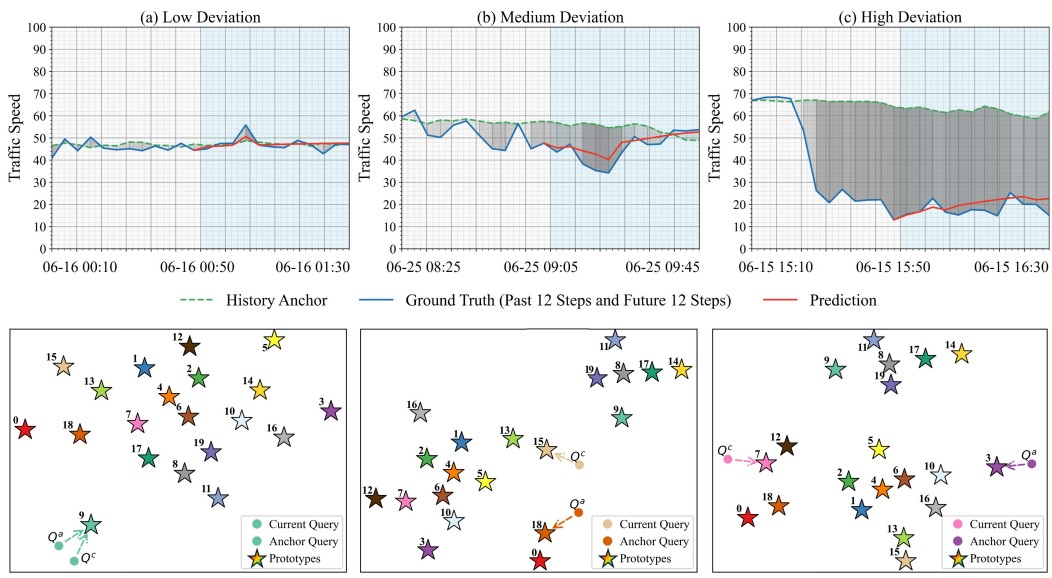

Figure 4: Visualization of predictions and query-prototype association under various deviation levels.

## 5.6 Case Study

**Performance across Deviation Levels.** To assess how ST-SSDL responds to varying levels of deviation, we visualize its forecasting behavior alongside the latent query-prototype association under low, medium, and high deviation scenarios in Figure 4. The gray shaded area on the left represents the past inputs, while the blue shaded part on the right denotes the future values. In the low deviation scenario, where the current sequence remains closely aligned with its historical anchor, ST-SSDL accurately follows the ground truth. The corresponding queries $Q^c$ and $Q^a$ are assigned to the same prototype, forming a compact structure in latent space. This behavior reflects the model's ability to preserve semantic consistency when deviation is minimal. In the medium deviation case, ST-SSDL routes $Q^c$ and $Q^a$ to different yet nearby prototypes. Although the current input deviates from its historical anchor, the prediction closely follows the ground truth. The proximity between their assigned prototypes indicates that the model preserves latent relational consistency. Under high deviation, the current input diverges substantially from the historical anchor. Despite this shift, the prediction remains aligned with the ground truth. ST-SSDL adapts by mapping $Q^c$ and $Q^a$ to well-separated prototypes, effectively capturing the altered spatio-temporal context in the latent space. These visualizations confirm that our method adaptively adjusts prototype associations and prediction outputs in accordance with the degree of deviation, validating its effectiveness in modeling dynamic deviation across spatio-temporal data.

**Prototype and Query in Latent Space.** Beyond individual cases, we examine the global organization along with the query associations of prototypes to further validate the latent space discretization by our prototypes. Figure 5 illustrates a PCA projection of prototypes, with four hundred sampled queries positioned in the same projected space based on their relative distances and colored by their assigned positive prototypes. For clarity, we visualize the seven most frequently selected prototypes. The visualization demonstrates that queries are grouped into compact cluster centered around their assigned prototypes, with clear separation between clusters. This visualization suggests that the latent space has been effectively discretized into semantically coherent regions. Moreover, such arrangement also reflects the

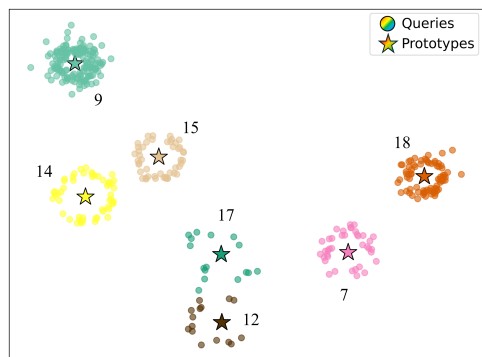

Figure 5: Visualization of prototype and query distribution in latent space.

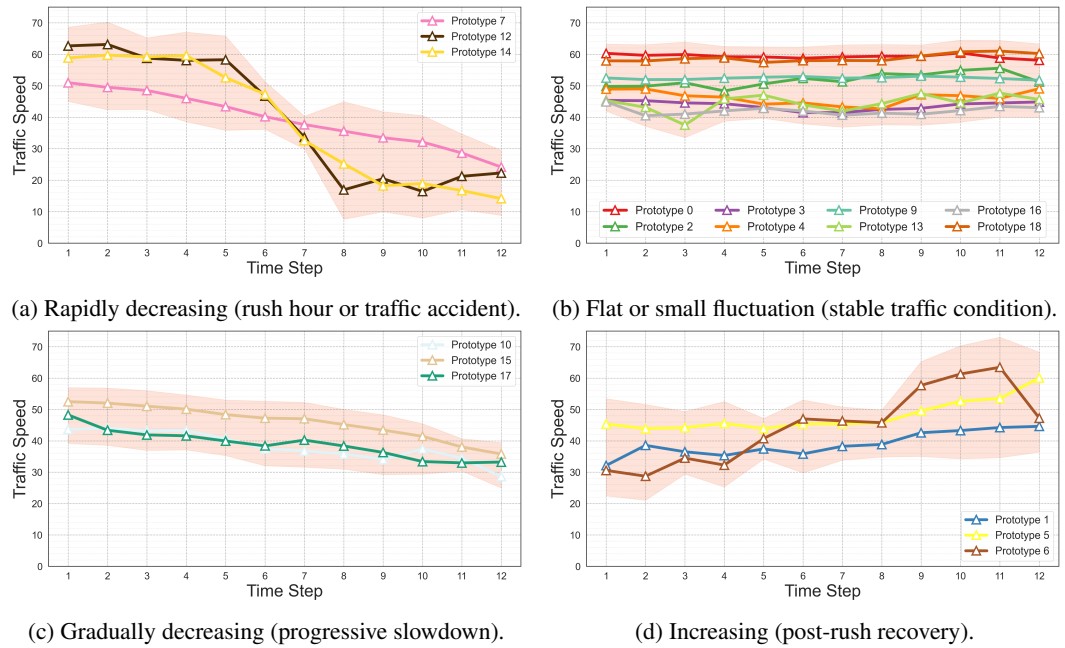

(a) Rapidly decreasing (rush hour or traffic accident).

(b) Flat or small fluctuation (stable traffic condition).

(c) Gradually decreasing (progressive slowdown).

(d) Increasing (post-rush recovery).

Figure 6: Prototype patterns recovered in the physical space on METRLA dataset. The shaded area indicates the standard deviation.

model's ability to capture diverse spatio-temporal patterns and organize them into representative prototypes, thereby supporting robust deviation modeling across varying input conditions.

**Prototype in Physical Space.** While Figure 5 visualizes the learned prototypes in latent space, we can also compute each prototype in physical space by collecting input sequences assigned to it, i.e., for prototype $P_k$, we collect input $X^c$ whose query $Q^c$ is assigned to $P_k$. Then we compute the pointwise average of these sequences to derive each prototype $P_k$. Figure 6 visualizes these patterns on METRLA and groups similar patterns together. Figure 6a exhibits a *rapidly decreasing* pattern, suggesting underlying peak-hour dynamics or incident-induced speed drops. Figure 6b displays *flat patterns with small fluctuations*. The abundance of prototypes in this category reflects stable operating states across diverse road segments. Figure 6c shows *gradually decreasing* modes with a smooth decline over 12 steps, consistent with progressive slowdowns in traffic speed. Figure 6d presents *increasing* paradigms aligned with post-peak recovery where speed returns toward free-flow conditions. These physical space diagrams summarize how the learned prototypes reveal recurrent traffic patterns in the real world.

## 6 Conclusion

This paper introduces ST-SSDL, a self-supervised framework that enhances spatio-temporal forecasting by explicitly modeling latent deviations. By using historical averages as context-aware anchors and discretizing the deviation space with learnable prototypes, ST-SSDL offers a structured approach to deviation-aware representation learning. The latent space is jointly optimized by contrastive and deviation losses, while avoiding reliance on explicit labels. Experiments on six benchmarks confirm its superior performance, and visualizations illustrate its ability to adapt under varying levels of deviation. Future directions include extending this framework to hierarchical prototype structures to further assess their adaptability and robustness.

## 7 Acknowledgement

This work was supported by JST CREST Grant Number JPMJCR21M2, JSPS KAKENHI Grant Number JP24K02996, JST BOOST Grant Number JPMJBS2418, and Toyota Motor Corporation.

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

# A Appendix

## A.1 Broader Impact

ST-SSDL has the potential to improve decision-making in domains such as traffic management, urban planning, and environmental monitoring, where accurate predictions can lead to reduced congestion, better resource allocation, and improved public safety. By introducing a self-supervised mechanism for modeling deviations, ST-SSDL further reduces reliance on labeled data and can adapt to changing conditions more robustly. However, like other data-driven forecasting systems, ST-SSDL relies on historical sensor data, which may reflect biases in urban infrastructure deployment. If such biases are not addressed, models may perform unevenly across regions, potentially reinforcing existing disparities.

## A.2 Summary of Notation

For reference, Table 5 summarizes the key notations and their descriptions in this paper.

Table 5: Notation Table

| Symbol | Description |
|---|---|
| $N$ | Number of nodes |
| $C$ | Number of input channels |
| $h$ | Dimension of latent representations $H$ |
| $d$ | Dimension of query $Q$ and prototypes $P$ |
| $T, T'$ | Length of input and output sequences, respectively |
| $X^c \in \mathbb{R}^{T \times N \times C}$ | Input sequence at current timestep |
| $X^a \in \mathbb{R}^{T \times N \times C}$ | Timestamp-aligned historical anchor sequence |
| $X^w \in \mathbb{R}^{S \times T^w \times N \times C}$ | Segmented training data by week |
| $\bar{X}^w \in \mathbb{R}^{T^w \times N \times C}$ | Weekly historical anchor (averaged) |
| $H^c, H^a \in \mathbb{R}^{N \times h}$ | Encoded latent states of current and historical input |
| $Q^c, Q^a \in \mathbb{R}^d$ | Query vectors derived from $H^c$ and $H^a$ |
| $\{\mathbf{P}_1, \ldots, \mathbf{P}_M\} \in \mathbb{R}^{M \times d}$ | Learnable prototype vectors in latent space |
| $\boldsymbol{\alpha}_i$ | Attention score between query and $i$-th prototype |
| $V$ | Attention-weighted representation from prototype pool |
| $\mathcal{P}^c, \mathcal{N}^c$ | Positive and Negative prototypes for $Q^c$ |
| $\mathcal{P}^a, \mathcal{N}^a$ | Positive and Negative prototypes for $Q^a$ |
| $\tilde{\mathcal{A}} \in \mathbb{R}^{N \times N}$ | Adaptive graph structure computed from latent states |
| $H'$ | Concatenated augmented hidden representation |
| $r_t, u_t$ | Reset and update gates in GCRU |
| $c_t$ | Candidate hidden state in GCRU |
| $\mathcal{L}_{\text{MAE}}$ | Mean Absolute Error loss for prediction accuracy |
| $\mathcal{L}_{\text{Con}}, \mathcal{L}_{\text{Dev}}$ | Contrastive loss and deviation loss |
| $\lambda_{\text{Con}}, \lambda_{\text{Dev}}$ | Loss weighting hyperparameters |
| $L$ | Number of GCRU layers |

## A.3 Detailed Dataset Description

Detailed description of the six benchmark datasets used in the experiments are provided in Table 6. The METRLA and PEMSBAY datasets provide traffic speed data collected from Los Angeles and the Bay Area, respectively. The PEMSD7(M), PEMS04, PEMS07, and PEMS08 datasets consist of traffic speed and flow records sourced from California's Performance Measurement System (PEMS)[3]. A brief introduction to PEMS: It is managed by the California Department of Transportation and provides data from 2001 to the present, collected from approximately 40,000 individual detectors deployed across California's freeway network in major metropolitan areas. These datasets offer a temporal resolution of 5 minutes, aggregated from original 30-second raw measurements and distributed in csv format.

---

[3] https://pems.dot.ca.gov/

Table 6: Detailed dataset descriptions.

| Dataset | #Sensors (N) | #Timesteps | Time Range | Frequency | Information |
|---------|--------------|------------|------------|-----------|-------------|
| METRLA | 207 | 34,272 | 03/2012 - 06/2012 | 5min | Traffic Speed |
| PEMSBAY | 325 | 52,116 | 01/2017 - 06/2017 | 5min | Traffic Speed |
| PEMSD7(M) | 228 | 12,672 | 05/2012 - 06/2012 | 5min | Traffic Speed |
| PEMS04 | 307 | 16,992 | 01/2018 - 02/2018 | 5min | Traffic Flow |
| PEMS07 | 883 | 28,224 | 05/2017 - 08/2017 | 5min | Traffic Flow |
| PEMS08 | 170 | 17,856 | 07/2016 - 08/2016 | 5min | Traffic Flow |

## A.4 Complete Pseudocode of ST-SSDL

---

**Algorithm 1** Spatio-Temporal Time Series Forecasting with Self-Supervised Deviation Learning

---

**Require:** Current Input $X^c \in \mathbb{R}^{T \times N \times C}$; corresponding historical anchor $X^a \in \mathbb{R}^{T \times N \times C}$; input length $T$; output length $T'$; number of variable $N$; number of prototypes $M$; dimension of latent representation $h$; hidden dimension of query and prototypes $d$;

**Ensure:** Forecasted sequence $\hat{X}^c$, loss $\mathcal{L}$

1: **// Encode input and anchor**
2: $H^c \leftarrow GCRU^{\text{enc}}(X^c)$                    $\triangleright H^c \in \mathbb{R}^{N \times h}$
3: $H^a \leftarrow GCRU^{\text{enc}}(X^a)$                    $\triangleright H^a \in \mathbb{R}^{N \times h}$
4: **// Project into query space**
5: $Q^c \leftarrow Linear(H^c)$                    $\triangleright Q^c \in \mathbb{R}^{N \times d}$
6: $Q^a \leftarrow Linear(H^a)$                    $\triangleright Q^a \in \mathbb{R}^{N \times d}$
7: **// Compute query-prototype attention score**
8: $\alpha^c \leftarrow \text{softmax}(Q^c \cdot \mathbf{P}^\top / \sqrt{d})$                    $\triangleright \alpha^c \in \mathbb{R}^{N \times M}$
9: $\alpha^a \leftarrow \text{softmax}(Q^a \cdot \mathbf{P}^\top / \sqrt{d})$                    $\triangleright \alpha^a \in \mathbb{R}^{N \times M}$
10: **// Retrieve top-2 prototypes**
11: $\mathcal{P}^c, \mathcal{N}^c \leftarrow \text{Top2}(\alpha^c)$
12: $\mathcal{P}^a, \mathcal{N}^a \leftarrow \text{Top2}(\alpha^a)$
13: **// Compute contrastive loss**
14: $\mathcal{L}_{\text{Con}} \leftarrow \max(\|\widetilde{\nabla}(Q^c) - \mathcal{P}^c\|_2^2 - \|\widetilde{\nabla}(Q^c) - \mathcal{N}^c\|_2^2 + \delta, 0)$
15: **// Compute deviation loss**
16: $d_q \leftarrow \|Q^c - Q^a\|_1$
17: $d_p \leftarrow \|\mathcal{P}^c - \mathcal{P}^a\|_1$
18: $\mathcal{L}_{\text{Dev}} \leftarrow \|\widetilde{\nabla}(d_q) - d_p\|_1$
19: **// Decode with adaptive adjacency**
20: $V^c \leftarrow \alpha^c \mathbf{P}, V^a \leftarrow \alpha^a \mathbf{P}$                    $\triangleright V^c \in \mathbb{R}^{N \times d}; V^a \in \mathbb{R}^{N \times d}$
21: $H' \leftarrow W[H^c | V^c | H^a | V^a] + b$                    $\triangleright H' \in \mathbb{R}^{N \times d}$
22: $\tilde{A} \leftarrow \text{Softmax}(\text{ReLU}(H' \cdot H'^\top))$                    $\triangleright \tilde{A} \in \mathbb{R}^{N \times N}$
23: $\hat{X}^c \leftarrow GCRU^{\text{dec}}([H^c, V^c], \tilde{A})$                    $\triangleright \hat{X}^c \in \mathbb{R}^{T' \times N \times C}$
24: **// Compute final loss**
25: $\mathcal{L}_{\text{MAE}} \leftarrow \frac{1}{NT'} \sum_{i=1}^{N} \sum_{t=1}^{T'} |\hat{X}_{i,t}^t - X_{i,t}^c|$
26: $\mathcal{L} \leftarrow \mathcal{L}_{\text{MAE}} + \lambda_{\text{Con}} \mathcal{L}_{\text{Con}} + \lambda_{\text{Dev}} \mathcal{L}_{\text{Dev}}$
27: **// Return the forecasting result**
28: **return** $\hat{X}^c$

---

## A.5 Detailed Explanation of *Lazy* Mode

The *lazy* mode refers to a collapse phenomenon in SSDL, where all query representations become overly similar and are mapped to the same prototype. This undermines the discriminative power of the model and limits its ability to capture diverse deviation patterns. In ST-SSDL, this issue arises when the gradient from self-supervised losses directly flows through both the query and prototype representations, allowing the model to minimize objectives trivially without learning meaningful structure.

To address this, we apply the stop-gradient operator $\widetilde{\nabla}(\cdot)$ to the query terms in both the contrastive loss $\mathcal{L}_{\text{Con}}$ and the deviation loss $\mathcal{L}_{\text{Dev}}$. In the contrastive loss:

$$\mathcal{L}_{\text{Con}} = \max\left(\|\widetilde{\nabla}(Q^c) - \mathcal{P}^c\|_2^2 - \|\widetilde{\nabla}(Q^c) - \mathcal{N}^c\|_2^2 + \delta, 0\right),$$

and in the deviation loss:

$$\mathcal{L}_{\text{Dev}} = \left\|\widetilde{\nabla}(\|Q^c - Q^a\|_1) - \|\mathcal{P}^c - \mathcal{P}^a\|_1\right\|_1,$$

the operator freezes the query during optimization, forcing the prototypes to adapt around the fixed distribution of query representations. This prevents the model from collapsing into trivial solutions (e.g., assigning all queries to a single prototype), which would otherwise minimize both loss terms without learning meaningful deviations.

By isolating the optimization to prototypes only, this strategy ensures that the latent space remains diverse and structured. It enables the prototype space to reflect semantic spatio-temporal patterns and maintain sufficient granularity for deviation quantification. Without this design, the model tends to fall into a shortcut solution, hence entering the *lazy* mode and severely degrading performance.

Besides theoretical analysis, we also empirically verify this phenomenon: removing the stop-gradient operation leads to single prototype to be chosen by all queries.

### A.6 Complete Efficiency Evaluation

We report the computational efficiency of ST-SSDL and all baseline models on the six datasets in Table 7, including the number of parameters, training time per epoch, and inference time. All of them are tested on an Intel(R) Xeon(R) Silver 4314 CPU @ 2.40GHz, 320G RAM computing server, equipped with NVIDIA RTX 5000 Ada Generation graphics cards. Moreover, we also provide a detailed descriptions of each baseline method as follows:

- **GRU** [7]: Gated Recurrent Unit, a type of recurrent neural network (RNN) that uses gating mechanisms (an update gate and a reset gate) to manage information flow , making it effective for sequence modeling tasks.

- **STGCN** [83]: spatio-temporal graph convolutional network, a deep learning model for traffic forecasting on spatio-temporal graphs based on graph convolution operation.

- **DCRNN** [44]: Diffusion convolutional recurrent neural network, a neural network for traffic forecasting using diffusion convolution and RNN.

- **GWNet** [77]: GWNet neural network, a CNN-based deep learning model for traffic forecasting using graph convolution layer and wavenet architecture.

- **MTGNN** [78]: Multivariate Time Series Graph Neural Network, a general framework for multivariate time series forecasting that automatically learns uni-directed relations among variables through a graph learning module, and captures spatial and temporal dependencies using novel mix-hop propagation and dilated inception layers, respectively.

- **AGCRN** [2]: Adaptive graph convolutional recurrent network for traffic forecasting, learning node-specific patterns through node adaptive parameter learning module and data adaptive graph generation module.

- **GTS** [62]: A model for multivariate time series forecasting that learns the underlying graph structure simultaneously with a Graph Neural Network (GNN) when this structure is not explicitly known. It achieves this by framing the problem as learning a probabilistic graph model, optimizing mean performance over a graph distribution that is parameterized by a neural network, enabling differentiable sampling of discrete graphs.

- **STNorm** [14]: Two normalization modules (temporal normalization and spatial normalization) are proposed to refine the high-frequency and local components of the raw data to help the model distinguish between time and space, respectively.

- **STID** [63]: STID tackles the issue of indistinguishable samples in traffic prediction by simply attaching spatial and temporal Identity information to the data. Using basic Multi-Layer Perceptrons (MLPs) with this approach, STID demonstrates that clearly identifying samples in space and time is crucial.

- **ST-WA** [8]: Spatio-temporal aware traffic time series forecasting model,turning spatio-temporal agnostic models into spatio-temporal aware models with encoding time series from different locations into stochastic variables.
- **STDN** [4]: STDN tackles complex traffic prediction by first building a dynamic graph and using spatio-temporal embeddings. Its core innovation is a module that decomposes traffic data into trend-cyclical and seasonal components for each node. An encoder-decoder then uses these components for prediction, achieving superior performance.

Table 7: Efficiency comparison on all 6 datasets.

| Model | METRLA ($N = 207$) | | | PEMSBAY ($N = 325$) | | | PEMSD7(M) ($N = 228$) | | |
|---|---|---|---|---|---|---|---|---|---|
| | #Params | Train | Infer | #Params | Train | Infer | #Params | Train | Infer |
| GRU | 126K | 71.96s | 3.21s | 126K | 172.32s | 7.62s | 126K | 27.96s | 1.30s |
| STGCN | 246K | 81.39s | 2.63s | 306K | 210.90s | 6.35s | 257K | 33.86s | 0.97s |
| DCRNN | 372K | 529.94s | 15.48s | 372K | 1071.18s | 33.03s | 372K | 854.15s | 30.72s |
| GWNet | 309K | 101.11s | 2.22s | 312K | 260.98s | 5.82s | 310K | 37.47s | 0.88s |
| MTGNN | 405K | 62.95s | 1.24s | 573K | 138.64s | 2.80s | 435K | 21.18s | 0.47s |
| AGCRN | 752K | 87.15s | 2.68s | 753K | 197.93s | 5.38s | 752K | 31.67s | 1.02s |
| GTS | 38377K | 190.48s | 4.86s | 58363K | 546.96s | 14.91s | 12158K | 54.12s | 1.70s |
| STNorm | 224K | 64.80s | 0.88s | 284K | 167.36s | 2.31s | 235K | 23.37s | 0.34s |
| STID | 118K | 12.06s | 0.75s | 122K | 19.25s | 0.62s | 118K | 4.71s | 0.21s |
| ST-WA | 375K | 113.90s | 7.41s | 447K | 284.30s | 19.25s | 388K | 41.21s | 2.78s |
| STDN | 5971K | 459.43s | 10.39s | 6273K | 1479.01s | 25.91s | 6025K | 170.05s | 4.37s |
| **ST-SSDL** | 498K | 71.84s | 8.65s | 498K | 199.60s | 32.40s | 181K | 49.66s | 6.64s |

| Model | PEMS04 ($N = 307$) | | | PEMS07 ($N = 883$) | | | PEMS08 ($N = 170$) | | |
|---|---|---|---|---|---|---|---|---|---|
| | #Params | Train | Infer | #Params | Train | Infer | #Params | Train | Infer |
| GRU | 126K | 50.87s | 2.30s | 126K | 249.13s | 11.05s | 126K | 31.23s | 1.36s |
| STGCN | 297K | 63.95s | 1.75s | 592K | 369.55s | 7.67s | 227K | 32.73s | 1.22s |
| DCRNN | 372K | 269.71s | 9.06s | 372K | 1330.26s | 48.46s | 372K | 211.56s | 6.61s |
| GWNet | 311K | 72.94s | 1.79s | 323K | 445.99s | 11.26s | 309K | 37.90s | 0.92s |
| MTGNN | 548K | 38.84s | 0.91s | 1368K | 212.79s | 5.13s | 353K | 28.48s | 0.51s |
| AGCRN | 749K | 58.13s | 1.69s | 755K | 342.73s | 9.89s | 150K | 38.44s | 1.18s |
| GTS | 16305K | 100.87s | 3.54s | 27088K | 1270.23s | 62.82s | 17136K | 69.07s | 1.79s |
| STNorm | 275K | 44.95s | 0.74s | 570K | 279.37s | 4.68s | 205K | 28.06s | 0.58s |
| STID | 121K | 5.74s | 0.26s | 140K | 12.32s | 0.64s | 117K | 5.42s | 0.24s |
| ST-WA | 436K | 76.69s | 5.70s | 786K | 678.93s | 60.99s | 353K | 47.88s | 3.80s |
| STDN | 6227K | 312.12s | 7.98s | 4480K | 693.32s | 32.56s | 5876K | 200.60s | 5.71s |
| **ST-SSDL** | 182K | 83.48s | 12.08s | 379K | 870.12s | 217.80s | 100K | 65.81s | 7.97s |

## A.7 Supplementary Case Study

To further understand the behavior of different models under various real-world conditions, we conduct a comprehensive visual comparison of forecasting results across six models: DCRNN, STGCN, AGCRN, MTGNN, STDN, and our proposed ST-SSDL on METRLA dataset. Figure 7, 8, 9, and 10 presents four representative cases: low deviation, medium deviation, high deviation, and partially missing values. Each case highlights specific challenges in spatio-temporal forecasting and demonstrates how different models respond under these conditions.

**Low-Deviation Scenario.** In this case, the current input remains highly consistent with its historical average. Most models are able to make accurate forecasts under this setting. However, we observe that AGCRN and STDN exhibit slight discrepancies in the prediction trajectory, possibly due to their reliance on static graph structures or insufficient temporal sensitivity. In contrast, ST-SSDL maintains precise alignment with the ground truth, demonstrating that it performs competitively even in standard settings.

**Medium-Deviation Scenario.** When moderate deviation occurs, certain models begin to show sensitivity to the changing input dynamics. DCRNN, STGCN, AGCRN, and MTGNN all produce predictions that visibly deviate from the ground truth, reflecting a struggle to generalize under moderate shifts. ST-SSDL, by contrast, adapts effectively to this condition and provides accurate

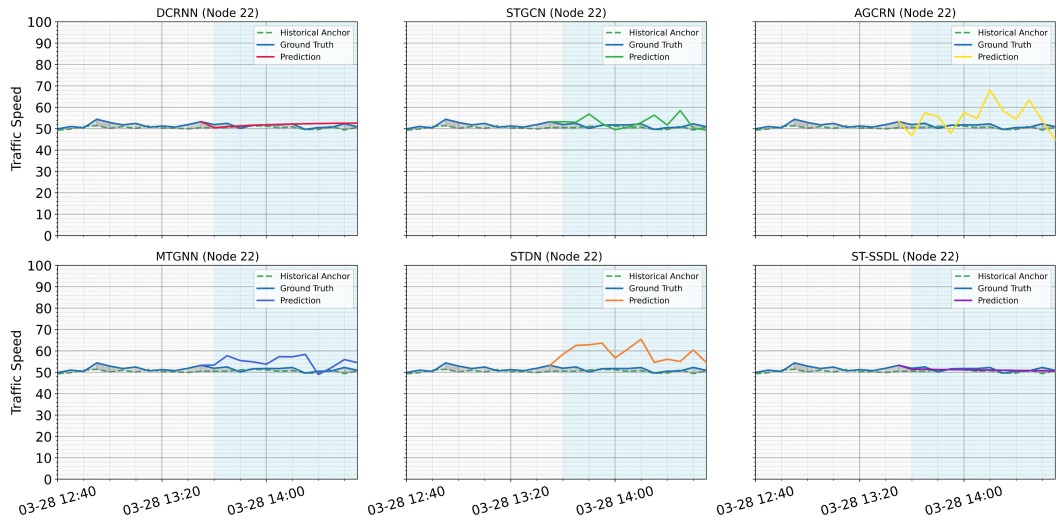

Figure 7: Prediction comparisons under **low deviation**.

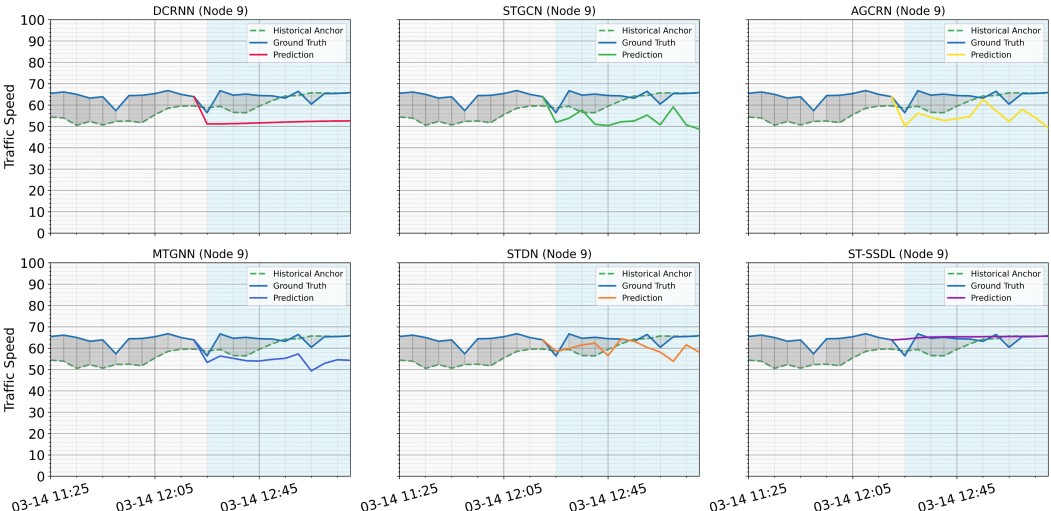

Figure 8: Prediction comparisons under **medium deviation**.

predictions. This supports the effectiveness of our self-supervised deviation modeling strategy, which guides the model to dynamically adjust its representation based on contextual shifts.

**High-Deviation Scenario.** In this more challenging case, a significant deviation is observed between current input and historical trends. All baseline models fail to capture this change and generate forecasts that diverge considerably from the actual values. ST-SSDL is the only model that maintains predictive accuracy, closely tracking the ground truth despite the abrupt change. This result underscores the necessity of modeling deviation in spatio-temporal forecasting and validates the core motivation of our proposed framework.

**Missing-Value Scenario.** Real-world data often contain missing or incomplete observations. In this setting, models such as DCRNN and STGCN suffer performance degradation due to the missing input points. ST-SSDL, however, continues to produce reliable predictions. This robustness may stem from its ability to interpret missing data as a form of deviation from historical norms, allowing it to handle imperfect inputs more effectively. This case highlights an additional advantage of deviation modeling: improved resilience to data corruption.

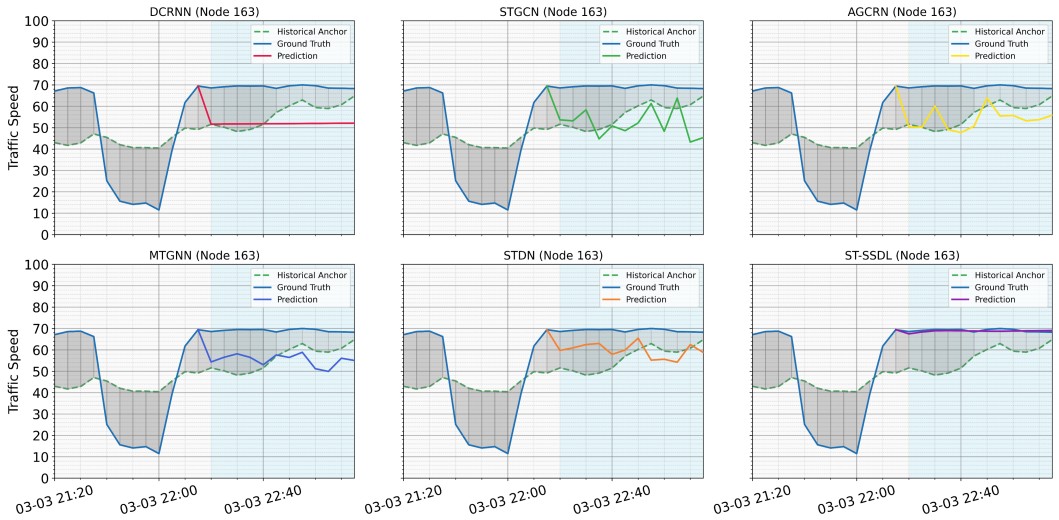

Figure 9: Prediction comparisons under **high deviation**.

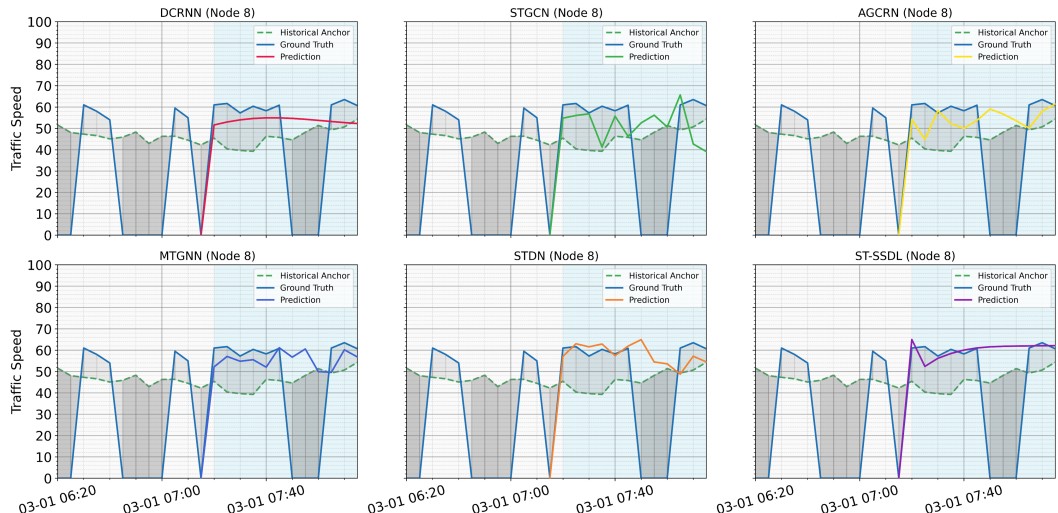

Figure 10: Prediction comparisons under **partially missing values**.

These visual analyses reinforce the motivation and effectiveness of our self-supervised deviation learning framework. ST-SSDL not only excels under large deviations but also maintains robustness in both common and imperfect scenarios.

