# OpenReview forum: "How Different from the Past? Spatio-Temporal Time Series Forecasting with Self-Supervised Deviation Learning"
_NeurIPS.cc/2025/Conference — NeurIPS 2025 poster_

### Official Review · Reviewer_CVvy · 2025-06-27

**Clarity:** 3
**Significance:** 2
**Originality:** 3
**Rating:** 3
**Confidence:** 3

**Summary:**

This submission introduces a spatio-temporal time series forecasting framework, ST-SSDL, which is based on a GCRU-based encoder-decoder and adopts Self-Supervised Deviation Learning (SSDL) to explicitly model dynamic deviations between current observations and historical trends using a self-supervised mechanism. The authors evaluated the proposed method on six traffic datasets. Quantitative benchmarks, ablation studies, efficiency metrics, and qualitative visualizations demonstrate the effectiveness of ST-SSDL.

**Questions:**

1. Since the paper focuses on spatio-temporal time series forecasting, which component in ST-SSDL specifically addresses spatial analysis?

2. How the anchors were computed in Section 4.1?

3. Have the authors considered applying ST-SSDL to other spatio-temporal domains such as weather, electricity, or medical time series?

4. Can you provide further semantic interpretation of the prototypes? Do they correspond to interpretable traffic states (e.g., rush hour, low traffic)?

**Ethical Concerns:**

["NO or VERY MINOR ethics concerns only"]

**Final Justification:**

As reviewer TcCg commented, the methodological contribution in this submission appears limited, in addition to ambiguous presentation. After reading all the reviews and discussions, I still have lingering concerns on the novelty of SSDL, which appears incremental and similar to previous prototype-based time series methods, learning prototypical representations or states to predict future states. I could not find any component of SSDL specifically designed to model spatial dependency, which might need to be considered based on the writing of the current abstract and introduction section. I will keep my current score for now.

**Limitations:**

The authors did not explicitly address the broader limitations or potential societal impacts. Some brief discussions on: i) Applicability to high-risk domains; and ii) Limitations in the anchoring mechanism; could be added.

**Quality:**

3

**Strengths And Weaknesses:**

Strengths
i) The paper focuses on explicit deviation modeling in spatio-temporal forecasting, and proposes a principled self-supervised approach to quantify and leverage deviations;
ii) The use of prototypes in latent space adds a structured, interpretable layer to the self-supervised learning pipeline;
iii) The proposed model is benchmarked on six well-known datasets and compared against a wide range of SOTA baselines. Ablation and efficiency studies are presented clearly;
iv) Prototype-based modeling provides interpretability in the latent space, and the visualizations (Figures 4 and 5) effectively demonstrate how the model adjusts to varying deviation levels;
v) The authors stated to release code and data, which supports reproducibility.

Weaknesses
i) The focus of the paper is the introduction of SSDL, which could be applicable in many dynamic prediction tasks. It is not clear how it may perform on different architectures besides GCRU;
ii) It is not clear how spatial relationships are captured or leveraged in ST-SSDL;
iii) The experiments are all based on traffic datasets. It is unclear how generalizable the proposed method is to other spatio-temporal domains (e.g., climate, mobility, energy);
iv) Although the model is lightweight in model parameters, inference latency is relatively high due to the GCRU backbone. This limits deployment in real-time applications;
v) There seem to be inconsistent notations: notation inconsistency: The authors used $X^t$ (e.g., first appeared in line 103 and throughout Section 4) to denote the recent input sequence $X_{\tau−T+1:\tau}$, but this superscript notation was never clearly defined and may be confused with a time index. Since $\tau$ was already used to indicate the current time, a more consistent and intuitive notation might be $X^\tau$ or simply $X_{t-T+1:t}$?
vi) In Section 5.6, Figure 4 was not really referred or discussed in text.

---

> ### Author Rebuttal · Authors · 2025-07-31
>
> # Response to Reviewer CVvy
>
> Thank you for your detailed comments! Those reviews are really helpful to improve our manuscripts. Below are detailed responses to questions.
>
> ## **Weakness**
>
> > Q1: The focus of the paper is the introduction of SSDL, which could be applicable in many dynamic prediction tasks. It is not clear how it may perform on different architectures besides GCRU;
>
> **A1**: We thank the reviewer for this insightful question. To verify its applicability beyond the GCRU architecture, we conduct experiments by creating two additional variants based on MLP and vanilla Transformer backbones.
>
> ||Metric (3/6/12 or Avg. )|MLP|MLP+SSDL|Transformer|Trans+SSDL| GCRU|Ours (GCRU+SSDL)|
> | ------ | ------------------------ | ------------------- | ------------------ | ------------------- | ---------- | ------------------ | ----------------- |
> |METRLA|MAE(3/6/12)| 3.11/3.81/4.93|**2.81/3.18/3.56**| 3.00/3.60/4.48|**2.68/3.02/3.40**|2.83/3.25/3.81|**2.62/2.98/3.38**|
> ||RMSE (3/6/12)|6.12/7.69/9.71|**5.56/6.66/7.62**| 5.93/7.26/9.04|**5.18/6.13/7.14**|5.38/6.46/7.74|**5.06/6.08/7.18**|
> ||MAPE (3/6/12)|8.44%/10.90%/14.95%|**7.64%/9.25%/10.81%**| 8.01%/10.24%/13.68% |**6.99%/8.32%/10.00%**|7.40%/8.87%/10.77% | **6.68%/8.04%/9.71%** |
> |PEMS04|MAE (Avg.)|26.03|**18.30**| 23.46| **18.26**|22.71|**18.15**|
> ||RMSE (Avg.)|40.15|**29.87**| 36.92|**30.42**|35.07|**29.74**|
> ||MAPE (Avg.)|17.88%|**12.35%**| 15.46%|**12.34%**|15.57%|**12.38%**|
>
> Our results show that integrating the SSDL module brings significant performance improvements to all architectures. This demonstrates SSDL's ability to effectively enhance diverse architectures for spatiotemporal forecasting.
> > Q2: It is not clear how spatial relationships are captured or leveraged in ST-SSDL;
>
> **A2**: We thank the reviewer for this question and would like to clarify this process. Spatial relationships in ST-SSDL are explicitly captured and leveraged through its **Graph Convolution Recurrent Unit (GCRU) backbone**, as detailed in Section 4.2 of our paper. **GCRU is a spatio-temporal encoder.** The core of this mechanism is the **graph convolution operation** together with gating mechanisms which is formally defined in Eq. (6) and Eq. (7). This mechanism enables the model to aggregate information from neighboring nodes at each time step, effectively modeling spatial dependencies based on the graph topology.
>
> > Q3: The experiments are all based on traffic datasets. It is unclear how generalizable the proposed method is to other spatio-temporal domains (e.g., climate, mobility, energy)
>
> **A3**: We thank the reviewer for this insightful suggestion. We agree that evaluating our proposed ST-SSDL on a more diverse range of spatio-temporal domains would provide a broader perspective on its generalization capabilities.
>
> Our primary focus on transportation stems from its status as a standard and challenging domain in spatio-temporal forecasting. The datasets we used are widely recognized benchmarks for validating many classical models, including **DCRNN (ICLR'18, 4900+ citations)** [1], **Graph WaveNet (IJCAI'19, 2900+ citations)** [2], **AGCRN (NeurIPS'20, 1700+ citations)** [3], **MTGNN (KDD'20, 2000+ citations)** [4], and **GTS (ICLR'21, 400+ citations)** [5].
>
> Given the rigorous evaluation on widely-adopted benchmarks, where our method demonstrates superior performance, we believe it can be readily extended to other spatio-temporal domains such as air quality prediction [6] and weather forecasting [7]. We will try to extend our method to more spatio-temporal domains in the future.
>
> > Q4: Although the model is lightweight in model parameters, inference latency is relatively high due to the GCRU backbone. This limits deployment in real-time applications
>
> **A4**: We thank the reviewer for highlighting this trade-off. We agree and **have clearly discussed** this limitation in the **Sec.5.4 Efficiency Study and Limitation section**. "The model’s inference speed is slightly affected by the GCRU architecture, though this overhead remains acceptable in most practical settings." While the inference latency is a little higher than some non-recurrent baselines due to the iterative GCRU architecture, we believe it is acceptable in most cases. We honestly present such trade-offs, yet believe our work is a valuable contribution in terms of state-of-the-art performance and parameter efficiency.
>
> > Q5: There seem to be inconsistent notations: notation inconsistency: The authors used (e.g., $X^t$ first appeared in line 103 and throughout Section 4) to denote the recent input sequence $X_{\tau-T+1:{\tau}^{'}}$, but this superscript notation was never clearly defined and may be confused with a time index. Since was already used to indicate the current time, a more consistent and intuitive notation might be $X^{\tau}$ or simply $X_{t-T+1:t}$?
>
> **A5**: We thank the reviewer for pointing out this potential notational ambiguity. Our intention for using the superscript 't' (e.g., in $X^t$, $H^t$, $Q^t$) is to create a consistent notation throughout the paper for all variables related to the **current** input, as opposed to the historical anchor ($X^a$, $H^a$, $Q^a$). Because $t$ is already refer to **current**, we use $\tau$ in subscript to denote the time index in Eq. (1) and Eq. (6).
>
> For clarity, actually we have included a comprehensive notation table in the appendix to clarify this convention. We acknowledge that defining this more explicitly in the main body of the paper would improve clarity, and we will revise the text accordingly in the final version to prevent any confusion with time index.
>
> > Q6: In Section 5.6, Figure 4 was not really referred or discussed in text.
>
> **A6**: We thank the reviewer for this comment. We acknowledge this mistake. We will revise line **304** from "high deviation scenarios." to "high deviation scenarios **as Figure 4**."
>
> ## **Questions**
>
> > Q7: How the anchors were computed in Section 4.1?
>
> **A7**: We would like to clarify the procedure for computing the historical anchors. As described in Section "**History as Self-Supervised Anchor**" in lines 120 to 125, the historical anchors are computed from the full training sequence. Specifically, the entire training set is partitioned into non-overlapping weekly segments. The historical anchor is then calculated by averaging the values at aligned time steps across all of these weekly segments. This process ensures that for any given input sequence, we can retrieve a timestamp-aligned historical average that serves as a robust, contextual baseline for deviation modeling.
>
> Note that the weekly anchoring strategy can be replaced by different ones such as daily/weekly/monthly average or even moving average according to applications.
>
> > Q8: Can you provide further semantic interpretation of the prototypes? Do they correspond to interpretable traffic states (e.g., rush hour, low traffic)?
>
> **A8**: We thank the reviewer for this meaningful suggestion. While Figure 5 visualizes clusters in latent space that each prototype surrounded by its assigned queries, we can also compute the prototype's pattern in the input space by following method:
>
> 1. Collect all input sequences assigned to $P_k$ (i.e., those $X_i$ whose query $Q_i$ assigned to $P_k$).
> 2. Compute the average of these sequences to form the prototype’s representative curve.
>
> Formally, it can be calculated as following equation:
> $$
> \text{Pattern}\_k = \frac{1}{|\{i: Q\_i \in P\_k\}|}\sum_{Q\_i \in P\_k} X\_i,
> $$
> where ($X_i$) denotes the input sequence whose query $(Q_i)$  assigned to prototype \($P_k$\).
>
> We then visualize the traffic speed patterns of prototypes and observe several distinct patterns on METR-LA dataset. Since NeurIPS does not allow image updates in rebuttal phase, we would like to describe some of the phenomena we discovered:
>
> - **Prototype 7, 12, 14**: Rapidly decreasing curves $\Rightarrow$ sudden‐drop events (i.e. traffic accident).
> - **Prototype 0, 2, 3, 4, 9, 13, 16, 18**: Flat curves at different levels $\Rightarrow$ stable traffic conditions with small fluctuation.
> - **Prototype 10, 15, 17**: Gradually decreasing curves $\Rightarrow$ morning or evening rush hour.
> - **Prototype 1, 5, 6,**: Increasing curves $\Rightarrow$ after morning or evening rush hour.
>
> Moreover, these representative patterns of each prototype also **align well with the visualization of Figure 4**, further demonstrating that these prototypes capture meaningful spatio-temporal patterns. We commit to including this meaningful visualization and analysis in the final version to strengthen interpretability.
>
> > Q9: Response to your low score at significance .
>
> **A9**: At last, we would like to emphasis the core contribution of our work. We believe the core contribution of our work is significant as it is the first to formally propose and address the critical, yet previously overlooked, problem of dynamic deviation in spatio-temporal forecasting. By proposing a novel self-supervised deviation learning method that explicitly models these deviations, we provide a new perspective for the field and demonstrate its effectiveness by achieving state-of-the-art performance on six widely used benchmarks.
>
> **Reference**:
>
> [1] Diffusion convolutional recurrent neural network: Data-driven traffic forecasting, ICLR, 2018.
>
> [2] Graph WaveNet for Deep Spatial-Temporal Graph Modeling, IJCAI, 2019.
>
> [3] Adaptive graph convolutional recurrent network for traffic forecasting, NeurIPS, 2020.
>
> [4] Connecting the dots: Multivariate time series forecasting with graph neural networks, SIGKDD, 2020.
>
> [5] Discrete Graph Structure Learning for Forecasting Multiple Time Series, ICLR, 2021.
>
> [6] A neural attention model for urban air quality inference: Learning the weights of monitoring stations, AAAI, 2018.
>
> [7] Weather forecasting using ensemble of spatial-temporal attention network and multi-layer perceptron. *Asia-Pacific Journal of Atmospheric Sciences*, *57*(3), 2020.

---

> > ### Comment · Reviewer_CVvy · 2025-08-04
> >
> > I appreciate the additional experiments and clarifications by the authors but have a couple of remaining questions:
> >
> > **Q2**: Thank you for your response. Since the spatial-temporal modeling capability of ST-SSDL primarily stems from GCRU, could the authors clarify the differences between SSDL and recent prototype-based approaches for multivariate time series modeling (e.g., S4M [1])?
> >
> > [1] S4M: S4 for multivariate time series forecasting with Missing values, ICLR, 2025.
> >
> > **Q7**: I am still unclear about how $X^a$ is computed. Could the authors provide further clarification?

---

> > > ### Author Response · Authors · 2025-08-04
> > > **Response to Questions**
> > >
> > > > **Q2**: Thank you for your response. Since the spatial-temporal modeling capability of ST-SSDL primarily stems from GCRU, could the authors clarify the differences between SSDL and recent prototype-based approaches for multivariate time series modeling (e.g., S4M [1])?
> > > >
> > > > [1] S4M: S4 for multivariate time series forecasting with Missing values, ICLR, 2025.
> > >
> > > **A2**: Thank you for the thoughtful feedback throughout this process and the follow-up questions. As we demonstrated in our response to your first question, the performance of GCRU is significantly enhanced by our SSDL method.
> > >
> > > || Metric (3/6/12 or Avg. ) | GCRU| Ours (GCRU+SSDL)  |
> > > | ------ | ---------- | ------------------ | ----------------- |
> > > |METRLA|MAE(3/6/12)|2.83/3.25/3.81|**2.62/2.98/3.38**|
> > > || RMSE (3/6/12)| 5.38/6.46/7.74| **5.06/6.08/7.18**|
> > > || MAPE (3/6/12)| 7.40%/8.87%/10.77% | **6.68%/8.04%/9.71%** |
> > > | PEMS04 | MAE (Avg.)| 22.71| **18.15**|
> > > ||RMSE (Avg.)| 35.07| **29.74**|
> > > ||MAPE (Avg.)| 15.57%| **12.38%**|
> > >
> > > While both ST-SSDL and S4M utilize prototype-based method, there are fundamental differences in their designs and objectives: ST-SSDL introduces prototypes to **discretize** the continuous latent space as a part of our approach for effectively capturing **dynamic deviations** between current inputs and historical patterns. To achieve this, the prototypes are optimized by two self-supervised losses, which are our core contributions: **Contrastive Loss ($L_{Con}$)** and **Deviation Loss ($L_{Dev}$)**. In contrast, S4M's goal is to **handle missing values**. Its prototypes serve as a **"bank"** to store and update patterns that are used to construct robust representations for missing data, which is a fundamentally different system in **role, assignment mechanism, and optimization strategy**.
> > >
> > > Moreover, for a detailed explanation of our core motivation and contribution, please kindly refer to our first response A1 to **Reviewer hTWY (Reviewer on the top)**.
> > >
> > > > **Q7**: I am still unclear about how $X^a$ is computed. Could the authors provide further clarification?
> > >
> > > **A7**: We thank the reviewer for the follow-up question and the opportunity to provide a more concrete explanation of how the historical anchor $X^a$ is computed.
> > >
> > > Here is a step-by-step example: consider **Sensor 0** at **Monday 00:00**.
> > >
> > > 1. **Retrieving History:** We first go through the **entire training dataset** and collect all the values of **Sensor 0** at every **Monday 00:00**.
> > > 2. **Computing Average:** We then compute the average of all these collected values. This is the historical value for **Sensor 0** at **Monday 00:00**.
> > >
> > > The same process is applied to each sensor at every time step within a week (e.g., for METR-LA with time interval of five minutes: Sensor 0 at Monday 00:00, Sensor 0 at Monday 00:05, …, Sensor $N$ at Sunday 23:50, Sensor $N$ at Sunday 23:55). The final result is a complete set of historical values for any node at any time in a week.
> > >
> > > Finally, for a given input $X^t$ (e.g. Sensor 0 at Monday 00:00~01:00, 12 steps), we can retrieve its historical anchor $X^a$ from this set by matching the node and time (12 historical values).
> > >
> > > It is worth noting that our anchoring mechanism is highly scalable. The weekly anchoring strategy can be replaced by different ones such as daily/weekly/monthly average or even moving average according to applications.
> > >
> > > ***If the above discussions address your concerns, we would greatly appreciate it if you could consider increasing the score.***

---

> > > > ### Comment · Reviewer_CVvy · 2025-08-05
> > > >
> > > > I truly appreciate the authors' further clarification. After reading all the reviews and discussions, I still have lingering concerns on the novelty of SSDL, which appears incremental and similar to previous prototype-based time series methods, learning prototypical representations or states to predict future states. I could not find any component of SSDL specifically designed to model spatial dependency, which might need to be considered based on the writing of the current abstract and introduction section.

---

> > > > > ### Author Response · Authors · 2025-08-05
> > > > >
> > > > > We sincerely thank you for your continued engagement and for providing further feedback on our work. We appreciate the opportunity to offer a final clarification on these important points.
> > > > >
> > > > > Regarding the novelty, we would like to clarify that our core contribution is not the use of prototypes, but rather **the modeling of deviation and the corresponding self-supervised learning paradigm we propose**. While prior works use prototypes to learn representative states, SSDL is uniquely designed to model the **relationship between states** (specifically, the deviation between a current input and its historical anchor). The novelty lies in our two self-supervised losses, which enforce a **"relative distance consistency"** and are purpose-built for this deviation-quantification modeling, a crucial aspect overlooked by prior work.
> > > > >
> > > > > Regarding spatial dependency, you are correct that the SSDL method itself is not designed to specifically model spatial dependency. However, we wish to clarify that SSDL is, by design, inherently **spatio-temporal-aware**. Its core function is to model the unique deviation for each specific node at each specific time step. This means the deviation it quantifies is not a spatial-static value. It is based on both the spatial context (i.e. different node) and the temporal information (i.e. different time step). This rich, dynamic deviation provides a powerful new signal for spatiotemporal forecasting task.
> > > > >
> > > > > Thank you once again for your thorough review. Your feedback has been invaluable in improving our manuscript.

---

### Official Review · Reviewer_TcCg · 2025-06-30

**Clarity:** 3
**Significance:** 2
**Originality:** 3
**Rating:** 4
**Confidence:** 2

**Summary:**

The paper introduces ST-SSDL, a spatio-temporal forecasting framework leveraging Self-Supervised Deviation Learning (SSDL) to capture dynamic deviations between current inputs and historical patterns. ST-SSDL anchors inputs to historical averages, discretizes the latent space with learnable prototypes, and optimizes using contrastive and deviation losses. Experiments on six benchmark datasets show it outperforms existing methods.

**Questions:**

1.	See Weakeness
2.	In Equation (4) and Equation (5), one loss uses MSE while the other uses MAE. What is the rationale behind this choice?

**Ethical Concerns:**

["NO or VERY MINOR ethics concerns only"]

**Final Justification:**

Thank you for your detailed response, which has addressed many of my concerns. However, I still think that this method shares similarities with certain VQ-VAE variants. While the application may differ, as you noted, the core approach remains quite similar. Consequently, the method's novelty appears limited. Nonetheless, I have revised my score to 4.

**Limitations:**

See above

**Quality:**

2

**Strengths And Weaknesses:**

Strengths:

1.	The motivation is clearly articulated: The paper identifies a critical gap in existing spatio-temporal models—ignoring dynamic deviations between current inputs and historical patterns.

2.	The proposed ST-SSDL mechanism, including historical anchoring and prototype-based discretization, provides strong interpretability. Visualizations (Fig. 4-5) demonstrate adaptive prototype assignments under varying deviation levels, enhancing model transparency.

Weaknesses:

1.	The baselines used for comparison are outdated. While the paper references more recent models, these are not included as baselines, e.g., the meta-learning framework HIMeta [13] from KDD 2024.

2.	The Query-Prototype mechanism closely resembles VQ-VAE’s codebook design, with attention-based soft assignment (Equation (3)) replacing hard quantization. The paper lacks a thorough discussion on how this differs from or improves upon VQ-VAE.

3.	The use of averaging to generate historical anchors may be overly simplistic. The paper does not explore or compare alternative dynamic anchoring strategies

4 . What is the difference between "dynamic deviations" and "distribution shift"? Can you cite literature to explain "dynamic deviations"?

---

> ### Author Rebuttal · Authors · 2025-07-31
>
> # Response to Reviewer TcCg
>
> We thank you for your valuable feedback and provide our responses below.
>
> ## **Weakness**
>
> > Q1: The baselines used for comparison are outdated. While the paper references more recent models, these are not included as baselines, e.g., the meta-learning framework HimNet[13] from KDD 2024.
>
> **A1:** Thank you for this suggestion. While STDN (AAAI'25) is our most recent baseline, we have now added HimNet (KDD'24) [1] and MegaCRN (AAAI'23) [2] to our comparisons. To ensure a direct and fair comparison, we ran the official code for these models locally and report their performance. For ST-SSDL, we report the average of multiple runs with different random seeds.
>
> * *Performance Comparison with MegaCRN and HimNet*
>
> | Model   | Metric  | METRLA (3/6/12)              | PEMSBAY (3/6/12)             | PEMSD7(M) (3/6/12)          | PEMS04 (Avg)         | PEMS07 (Avg)         | PEMS08 (Avg)         |
> |--------------|---------------|------------------------------|------------------------------|-----------------------------|----------------------|----------------------|----------------------|
> | **MegaCRN**   | **MAE** | 2.62 / 3.01 / 3.48           | 1.28 / 1.60 / 1.89           | 2.05 / 2.67 / 3.27          | 18.69               | 19.70               | 14.83               |
> |   | **RMSE** | 5.04 / 6.13 / 7.31           | 2.71 / 3.68 / 4.45           | 3.88 / 5.34 / 6.68          | 30.54               | 32.72               | 23.92               |
> |             | **MAPE** | 6.68% / 8.18% / 9.98%        | 2.67% / 3.59% / 4.48%        | **4.75%** / 6.59% / 8.50%   | 12.71%              | 8.29%               | 9.64%               |
> | **HimNet**    | **MAE** | **2.61** / **2.96** / 3.38   | **1.27** / **1.58** / **1.86** | 2.06 / 2.64 / 3.19          | 18.24               | 19.30               | **13.56**           |
> |    | **RMSE** | **5.04** / 6.09 / 7.26       | 2.69 / **3.63** / **4.37**   | 3.94 / 5.32 / 6.57          | 30.12               | 32.77               | 23.36               |
> |             | **MAPE** | 6.71% / 8.15% / 9.97%        | 2.67% / 3.58% / 4.42%        | 4.87% / 6.64% / 8.30%       | **12.13%**          | **8.05%**           | **9.02%**           |
> | **ST-SSDL (Mean)** | **MAE** | 2.62 / 2.98 / **3.38**       | 1.28 / 1.59 / 1.88           | **2.04** / **2.59** / **3.09** | **18.15**          | **19.23**           | 13.89               |
> |      | **RMSE** | 5.06 / **6.08** / **7.18**   | **2.69** / 3.65 / 4.39       | **3.87** / **5.22** / **6.39** | **29.74**          | **32.67**           | **23.17**           |
> |      | **MAPE** | **6.68%** / **8.04%** / **9.71%** | **2.66%** / **4.39** / **4.41%** | 4.81% / **6.50%** / **8.06%** | 12.38%              | 8.11%               | 9.15%               |
>
>
> * *Efficiency Comparison with HimNet*
>
> || METRLA (#Para/Train/Infer) | PEMS07 (#Para/Train/Infer) | PEMS08 (#Para/Train/Infer) |
> | ------- | -------------------------- | -------------------------- | -------------------------- |
> | HimNet  | 1251K / 157.50s /17.29s| 1262K / 1354.37s / 201.77s | 2070K / 101.01s / 11.41s|
> | ST-SSDL | 498K / 71.84s / 8.65s| 379K / 870.12s / 217.80s| 100K /65.81s / 7.97s|
>
> Our proposed ST-SSDL outperforms MegaCRN on all six datasets and reaches a better or comparable performance against HimNet. Besides performance, ST-SSDL only uses 5% to 40 % parameters and runs 45% to 65% training time compared to HimNet.
>
> > Q2: The Query-Prototype mechanism closely resembles VQ-VAE’s codebook design, with attention-based soft assignment (Equation (3)) replacing hard quantization. The paper lacks a thorough discussion on how this differs from or improves upon VQ-VAE.
>
> **A2:** We sincerely thank the reviewer for this  insightful comment. While there is a visual resemblance in using a set of learnable vectors, we would like to clarify two **fundamental differences** with VQ-VAE [3] as follows:
>
> 1. Goal:
>    Our first fundamental difference lies in the core **goal**. Our method is purposed to **quantify the deviation** between current input and its historical anchor. In this context, the prototypes act as a structured reference grid to measure these dynamic changes. This directly contrasts with VQ-VAE, whose primary goal is **data compression and generation**. It aims to represent an input via a single discrete index from its codebook for efficient, generative purposes, a goal entirely different from ours.
> 2. Assignment Mechanism:
> To measure the dynamic deviations, our model requires a rich representation. Our **soft, cross attention-based** approach in Eq. (3) provides a **smooth and differentiable** landscape, enabling stable and effective optimization of our deviation-aware objectives. Conversely, VQ-VAE uses **hard quantization**, selecting only the single closest codebook vector. While effective for compression, this hard assignment mechanism is **non-differentiable** and can be unstable in training process.
>
> We appreciate the reviewer for prompting this comparison and will add a discussion to the revised manuscript to clarify these fundamental distinctions.
> > Q3: The use of averaging to generate historical anchors may be overly simplistic. The paper does not explore or compare alternative dynamic anchoring strategies.
>
> **A3:** Thank you for this insightful point. Actually, our anchoring mechanism is inspired by $HA$ (Historical Average) [4,5,6] ,which is always an important baseline almost for every application in time series forecasting domain. Especially for most spatio-temporal domains (data has clear periodic cycles), we consider historical weekly pattern is a simple yet effective anchoring strategy. Of course, we agree that there should be different anchoring strategies such as daily/weekly/monthly average or even moving average according to applications. Our core contribution, the deviation learning, should be recognized regardless of the anchoring strategy employed. We will add such kind of discussion in Sec.4.1 in the future version.
>
> > Q4: What is the difference between "dynamic deviations" and "distribution shift"? Can you cite literature to explain "dynamic deviations"?
>
> **A4:** We thank the reviewer for this comment. We would like to clarify the distinction between "dynamic deviations"and the well-known problem of "distribution shift".
> **Dynamic Deviation** refers to the changing difference **between a current observation and its corresponding historical average (anchor)**. The term "dynamic" emphasizes that the magnitude of this deviation **varies across different nodes and time points**, as we illustrate in the Introduction section. Our work proposes a method to explicitly model this varying signal to improve forecasting accuracy.
>
> **Distribution Shift**, in contrast, is a classical problem concerning a change in the underlying data distribution $P(X,Y)$ **between the training and testing sets**. This is a well-established challenge in machine learning, where a model trained on one statistical distribution performs poorly when evaluated on another.
>
> Regarding the request for literature on "dynamic deviations," we respectfully clarify that, to the best of our knowledge, our work is the **first to propose deviation learning based on this concept** in spatio-temporal forecasting domain. Therefore, the idea as presented is a novel contribution of our paper, aiming to establish it as a critical, yet previously overlooked, aspect of spatio-temporal modeling. We will include this discussion in the final manuscript.
>
> ## **Questions**
>
> > Q5: In Equation (4) and Equation (5), one loss uses MSE while the other uses MAE. What is the rationale behind this choice?
>
> **A5:** We thank the reviewer for this question, as our loss functions are deliberately chosen for their distinct objectives. We would like to clarify that Eq. (4), $L_{Con}$ we adopt a **variant of the Triplet Loss** [7] (squared L2 norm is typically used),  as noted in the **line 155** of our manuscript. We use squared L2 norm following its original equation. Conversely, for $L_{Dev}$ , we choose the Mean Absolute Error (MAE) for its robustness. Since this "distance-of-distances" manner can encounter large discrepancies in cases of high deviation, using an MSE would square these differences and create unstable gradients. The MAE loss provides a more stable training signal, preventing the model from being overly perturbed by extreme cases.
>
> > Q6: Response to your low score at significance.
>
> **A6**: At last, we would like to emphasis the core contribution of our work. We believe the core contribution of our work is significant as it is the first to formally propose and address the critical, yet previously overlooked, problem of dynamic deviation in spatio-temporal forecasting. By proposing a novel self-supervised deviation learning method that explicitly models these deviations, we provide a new perspective for the field and demonstrate its effectiveness by achieving state-of-the-art performance on six widely used benchmarks.
>
> **Reference**:
>
> [1] Heterogeneity-informed meta-parameter learning for spatiotemporal time series forecasting, SIGKDD, 2024.
>
> [2] Spatio-temporal meta-graph learning for traffic forecasting, AAAI, 2023.
>
> [3] Neural discrete representation learning, NeurIPS, 2017.
>
> [4] Pre-training enhanced spatial-temporal graph neural network for multivariate time series forecasting, SIGKDD, 2022.
>
> [5] Dl-traff: Survey and benchmark of deep learning models for urban traffic prediction, CIKM, 2021.
>
> [6] Z-GCNETs: Time zigzags at graph convolutional networks for time series forecasting, ICML, 2021.
>
> [7] Facenet: A unified embedding for face recognition and clustering, CVPR, 2015.

---

> ### Author Response · Authors · 2025-08-03
> **Response to Acknowledgment**
>
> Thank you for acknowledging our rebuttal. We truly hope our detailed response is helpful in clarifying our work and addressing your concerns. We would welcome any further discussion you might suggest. **If you find that we have addressed your concerns, could you kindly consider increasing your score?**

---

> > ### Comment · Reviewer_TcCg · 2025-08-04
> >
> > Thank you for addressing the main concern. Bumping the score to 4.

---

> > > ### Author Response · Authors · 2025-08-04
> > >
> > > It is great to know that our explanations and clarifications have made a positive impact on the paper. We deeply appreciate your acknowledgement and support. Thank you again for taking the time to help improve our work.

---

### Official Review · Reviewer_DAZY · 2025-07-03

**Clarity:** 4
**Significance:** 3
**Originality:** 3
**Rating:** 5
**Confidence:** 2

**Summary:**

This paper proposes ST-SSDL, a framework for forecasting spatio-temporal time series by explicitly modeling how current patterns deviate from historical trends. ST-SSDL uses a self-supervised approach: it compares current input data to its historical average, then represents both in a structured latent space using learnable prototypes that capture typical patterns. Two additional training objectives help the model quantify and organize these deviations without needing labeled data. Experiments on six benchmark datasets show that ST-SSDL consistently outperforms existing methods in prediction accuracy and adapts well to changing conditions.

**Questions:**

1. The paper assumes historical averages are reliable anchors, but real-world data often contains outliers or non-stationary trends (e.g., sudden traffic disruptions). Could ST-SSDL integrate robust statistics or outlier detection to improve deviation modeling in such cases?
2. The experiments focus on traffic data with stable periodic patterns. How might ST-SSDL need to be modified for domains with irregular or aperiodic deviations (e.g., disease spread or social media activity)?
3. Have the authors analyzed whether learned prototypes correspond to interpretable or human-understandable patterns?

**Ethical Concerns:**

["NO or VERY MINOR ethics concerns only"]

**Final Justification:**

The rebuttal effectively addressed my main concerns, including the robustness of the anchoring mechanism, applicability to irregular spatio-temporal domains, and interpretability of learned prototypes. I found the additional explanations and planned improvements to be reasonable and constructive. While some limitations remain, such as reliance on traffic data and potential generalizability issues, the paper is technically solid, presents a novel self-supervised approach, and demonstrates consistent empirical improvements. Therefore, I am keeping my score of 5.

**Limitations:**

yes

**Quality:**

3

**Strengths And Weaknesses:**

The paper is well-written and methodologically rigorous, introducing a novel self-supervised framework (ST-SSDL) that effectively models deviations in spatio-temporal data. It demonstrates strong empirical results across six benchmarks, supported by thorough ablation studies and clear visualizations, while maintaining reproducibility through open-sourced code and detailed experimental setups.
While technically sound, the paper has several limitations: (1) its reliance on historical averages as anchors isn't tested against noisy or non-stationary data (e.g., sudden traffic disruptions); (2) the 20 prototypes, while empirically chosen, lack interpretability as real-world patterns; (3) evaluation is confined to periodic traffic data, leaving generalization to irregular domains (e.g., epidemiology) unverified;

---

> ### Author Rebuttal · Authors · 2025-07-31
>
> # Response to Reviewer DAZY
>
> We are very grateful for your positive recognition and would like to thank you for the thoughtful comments. We respond to your feedback below.
>
> ## **Questions**
>
> > Q1:  The paper assumes historical averages are reliable anchors, but real-world data often contains outliers or non-stationary trends (e.g., sudden traffic disruptions). Could ST-SSDL integrate robust statistics or outlier detection to improve deviation modeling in such cases?
>
> **A1:** We appreciate the reviewer’s concern. As described in Section 4.1, our anchor is computed as the average over **the whole** training set. This global average inherently **smooths out** occasional outliers: any single disruption (e.g., an accident-induced outlier) contributes negligibly when aggregated with the full history.
>
> Note that, the anchoring strategy is a **flexible** component of our framework. For our task, we consider historical average as a simple yet effective strategy. Of course, robust statistics or outlier detection can be incorporated for better performance. We will add this in future work especially for more noisy applications.
>
> > Q2: The experiments focus on traffic data with stable periodic patterns. How might ST-SSDL need to be modified for domains with irregular or aperiodic deviations (e.g., disease spread or social media activity)?
>
> **A2:** Thank you for this insightful point. We would like to respond in two aspects:
>
> For aperiodic data, actually, our anchoring mechanism is inspired by $HA$ (Historical Average) [1,2,3] ,which is always an important baseline almost for every application in time series forecasting domain. For most spatio-temporal domains like transportation, climate, and energy, they are with clear periodic cycles (**daily, weekly, monthly**). Here we utilize historical weekly pattern as a **simple yet effective** anchoring strategy. As you suggested, we will try to incorporate more robust anchoring strategies to domains with aperiodic data in the future.
>
> For irregular deviations, while many spatio-temporal domains like transportation and climate exhibit clear periodic cycles, they also contains significant irregular deviations caused by **accidents and other sudden events (i.e. traffic control)**, which presents a major challenge for forecasting. By explicitly modeling the deviation between current inputs and historical patterns, our model can adapt to irregular disruptions. As we demonstrate in **Figure 4(c)** in **Sec. 5.6 Case Study**, ST-SSDL effectively handles high-deviation scenarios by mapping the current input and its historical anchor to well-separated prototypes in the latent space, thus capturing the potential irregular event.
>
> We will include such discussion in the final manuscript.
>
> > Q3: Have the authors analyzed whether learned prototypes correspond to interpretable or human-understandable patterns?
>
> **A3:** We thank the reviewer for this meaningful suggestion. While Figure 5 visualizes clusters in latent space that each prototype surrounded by its assigned queries, we can also compute the prototype's pattern in the input space by following method:
>
> 1. Collect all input sequences assigned to $P_k$ (i.e., those $X_i$ whose query $Q_i$ assigned to $P_k$).
> 2. Compute the average of these sequences to form the prototype’s representative curve.
>
> Formally, it can be calculated as following equation:
> $$
> \text{Pattern}\_k = \frac{1}{|\{i: Q\_i \in P\_k\}|}\sum_{Q\_i \in P\_k} X\_i,
> $$
> where ($X_i$) denotes the input sequence whose query $(Q_i)$  assigned to prototype \($P_k$\).
>
> We then visualize the traffic speed patterns of prototypes and observe several distinct patterns on METR-LA dataset. Since NeurIPS does not allow image updates in rebuttal phase, we would like to describe some of the phenomena we discovered:
>
> - **Prototype 7, 12, 14**: Rapidly decreasing curves $\Rightarrow$ sudden‐drop events (i.e. traffic accident).
> - **Prototype 0, 2, 3, 4, 9, 13, 16, 18**: Flat curves at different levels $\Rightarrow$ stable traffic conditions with small fluctuation.
> - **Prototype 10, 15, 17**: Gradually decreasing curves $\Rightarrow$ morning or evening rush hour.
> - **Prototype 1, 5, 6,**: Increasing curves $\Rightarrow$ after morning or evening rush hour.
>
> Moreover, these representative patterns of each prototype also **align well with the visualization of Figure 4**, further demonstrating that these prototypes capture meaningful spatio-temporal patterns. We commit to including this meaningful visualization and analysis in the final version to strengthen interpretability.
>
> **Reference**:
>
> [1] Dl-traff: Survey and benchmark of deep learning models for urban traffic prediction, CIKM, 2021.
>
> [2] Pre-training enhanced spatial-temporal graph neural network for multivariate time series forecasting, SIGKDD, 2022.
>
> [3] Z-GCNETs: Time zigzags at graph convolutional networks for time series forecasting, ICML, 2021.

---

> ### Comment · Reviewer_DAZY · 2025-08-04
>
> Thank you for the detailed rebuttal. I appreciate the clarifications, especially regarding the robustness of the anchoring mechanism, the generalizability to irregular domains, and the interpretability of the prototypes. These responses address my main concerns, and I will keep my score of 5.

---

> ### Author Response · Authors · 2025-08-05
>
> We are particularly grateful for your recognition and insightful feedback on our paper. Thank you again for your time and continued support in our work.

---

### Official Review · Reviewer_hTWY · 2025-07-05

**Clarity:** 2
**Significance:** 4
**Originality:** 3
**Rating:** 4
**Confidence:** 2

**Summary:**

The paper introduces a method to consider past events through a latent space that is then integrated into the learning process. The authors present experimental data that they claim supports their insight.

**Questions:**

into  S non-overlapping weekly ->why weekly? Shouldn´t this be application dependent?

. Experiment statistical significance-.> I do not understand tthat reply, can´t you use a fixed number of seeds?

- with dimensions h set to 128, 64,
242 or 32, depending on the dataset."
do you have a rule for this choice?

figure 5: can you move back from the latent space to the original data?

**Ethical Concerns:**

["NO or VERY MINOR ethics concerns only"]

**Limitations:**

...

**Paper Formatting Concerns:**

...

**Quality:**

3

**Strengths And Weaknesses:**

Strengths:

the idea that you can take advantage of re-occurring irregularities makes good sense. For example, a car mightperform differently in a highway or in busy city  street.
\\the evaluation seems to support these claims

Weaknesses
The authors present the work in a very abstract way. An example would have make things  clear, and would motivate the work.
Fig 1: I couldn;t quite understand.
The results are extensive but for a very specific type of domains

---

> ### Author Rebuttal · Authors · 2025-07-31
>
> # Response to Reviewer hTWY:
>
> We would first like to thank you for the positive comments and valuable feedback. We respond to your comments and questions below.
>
> ## **Weakness**
>
> > Q1: Let's start from the first weakness. The authors present the work in a very abstract way. An example would have make things clear, and would motivate the work. Fig 1: I couldn;t quite understand.
>
> **A1:** The entire motivation behind Figure 1 is as follows:
>
> (1) **History as anchor**. Historical Average [1,2,3] is always a simple yet effective approach for canonical tasks of time series forecasting. This motivates us to utilize historical average as an anchor (a reference) for each input to boost the forecasting performance. $D_1$ and $D_2$ in Figure 1(a) denotes the deviation (distance) between the current input and its anchor in physical space. The "?" in Figure 1(b) represents the corresponding deviation (distance) in latent space. Actually, they should be clearly detoned as $\widetilde D_1$ and $\widetilde D_2$.
>
> (2) **How to learn/quantify $\widetilde D_1$ and $\widetilde D_2$?** $D_1$ and $D_2$ are quite easy-to-calculate in physical space. In Figure 1(a), $D_1$ is approximately equal to 40 and $D_2$ is approximately equal to 20. However, it is never easy to accurately quantify $\widetilde D_1$ $\widetilde D_2$ in continuous latent space. The two “?” marks also conveys the core challenge of our study: how much should $\widetilde D_1$ $\widetilde D_2$ be equal to? i.e., $\widetilde D_1$ = ? and $\widetilde D_2$ = ?.
>
> To address the challenge above, we come up with our core idea: **relative distance consistency**. **The current-history pairs that are close (far) in the physical space should remain close (far) in latent space**. i.e. $D_1$ > $D_2$ $\Rightarrow$ $\widetilde D_1$ > $\widetilde D_2$.
>
> To implement this core idea, first, we propose to **discretize** the continuous latent space with a set of learnable **prototypes**, where a contrastive loss $L_{Con}$ (Eq. (4)) is utilized to guide the learning (i.e., discretizing) process.
>
> Next, we map the latent representation to its nearest prototype through cross attention, so as to take each prototype as the proxy of the latent representation. $P^t$ and $P^a$ in Eq. (5) are the proxy prototypes for the current input $X^t$ and its historical anchor $X^a$. Then their distance $| P^t - P^a |$ can **approximately represent** $\widetilde D$. By stopping the gradient of $Q^t$ and $Q^a$ ($\widetilde{\nabla}$), $|\widetilde{\nabla}(Q^t-Q^a)|$ can be taken as an approximate of physical-space distance $D$.
>
> Finally, the self-supervised deviation loss $L_{Dev}$ (Eq. (5)) is proposed to minimize $D- \widetilde D$ to implement the **relative distance consistency**, i.e., $D\uparrow \to\widetilde D\uparrow$，$D\downarrow\to\widetilde D\downarrow$. Please also kindly refer to Figure 2 for the whole implementation.
>
> As a successful learning result, we kindly ask you to refer to **Figure 4** in **Case Study (Sec. 5.6)**:
>
> - **Low deviation**  – current and historical curves nearly overlap; their latent representations are also close and select the *same* prototype.
>
> - **Medium deviation** – a medium gap in the physical space corresponding to a moderate latent distance; the two queries are assigned to two *nearby* prototypes.
>
> - **High deviation** – a large physical gap leads to widely separated latent representations and two *distant* prototypes.
>
> In the final version, we will make this logical flow clearer by revising the explanation of Figure 1 and explicitly stating the principle of relative distance consistency in the Introduction. We hope this could resolve the concerns and making our motivation clearer.
>
> > Q2: The results are extensive but for a very specific type of domains.
>
> We thank the reviewer for this insightful suggestion. We agree that evaluating our proposed ST-SSDL on a more diverse range of spatio-temporal domains would provide a broader perspective on its generalization capabilities.
>
> Our primary focus on transportation stems from its status as a standard and challenging domain in spatio-temporal forecasting. The datasets we used are widely recognized benchmarks for validating many classical models, including **DCRNN (ICLR'18, 4900+ citations)** [4], **Graph WaveNet (IJCAI'19, 2900+ citations)** [5], **AGCRN (NeurIPS'20, 1700+ citations)** [6], **MTGNN (KDD'20, 2000+ citations)** [7], and **GTS (ICLR'21, 400+ citations)** [8].
>
> Given the rigorous evaluation on widely-adopted benchmarks, where our method demonstrates superior performance, we believe it can be readily extended to other spatio-temporal domains such as air quality prediction [9] and weather forecasting [10]. We will try to extend our method to more spatio-temporal domains in the future.
>
> ## Question
>
> > Q3: into S non-overlapping weekly ->why weekly? Shouldn´t this be application dependent?
>
> **A3:** Thank you for highlighting this. Yes, your suggestion is absolutely correct, this partition length is **application-dependent**. We adopted weekly partitions due to the clear and widely recognized weekly periodicity in transportation data. As stated in **Sec. 4.1**, *"These anchors summarize recurring spatio-temporal patterns and serve as references for deviation modeling".* For domains with different temporal patterns, it could be adjusted accordingly. Importantly, ST-SSDL's core design does not inherently depend on weekly periodicity, making such adjustments feasible. We will clearly note this point in Sec. 4.1 in the final version.
>
> > Q4:  Experiment statistical significance-.> I do not understand tthat reply, can´t you use a fixed number of seeds?
>
> **A4:** Thank you for pointing this out. We have now conducted additional experiments with multiple random seeds (e.g., 5 runs per setting) on all six datasets. The results, including **mean and standard deviation**, demonstrate that ST-SSDL's performance gains over baseline models remain consistent and statistically significant. We will report these new results in the final version.
>
> | Dataset | Horizon | MAE | RMSE | MAPE (%) |
> | :--- | :--- | :---: | :---: | :---: |
> | **METRLA** | 15 min (Step 3) |2.62±0.02|5.06±0.04|6.68±0.06 |
> | | 30 min (Step 6) | 2.98±0.02 |6.08±0.04|8.04±0.09|
> | | 60 min (Step 12)| 3.38±0.02 |7.18±0.03|9.71±0.11|
> | **PEMSBAY** | 15 min (Step 3) |1.28±0.02|2.69±0.06|2.66±0.06 |
> | | 30 min (Step 6) | 1.59±0.03 |3.65±0.07|3.57±0.09|
> | | 60 min (Step 12)| 1.88±0.03 |4.39±0.06|4.41±0.08|
> | **PEMSD7(M)**| 15 min (Step 3) |2.04±0.01|3.87±0.03|4.81±0.05 |
> | | 30 min (Step 6) |2.59±0.01|5.22±0.03|6.50±0.07|
> | | 60 min (Step 12)|3.09±0.01|6.39±0.02|8.06±0.06|
> | **PEMS04** |Average|18.15±0.07|29.74±0.08|12.38±0.10|
> | **PEMS07** |Average|19.23±0.06|32.67±0.12|8.11±0.08|
> | **PEMS08** |Average|13.89±0.09|23.17±0.12|9.15±0.11|
>
>
> > Q5: with dimensions h set to 128, 64, 242 or 32, depending on the dataset." do you have a rule for this choice?
>
> **A5:** Actually, we do not follow a rigid rule. Instead, we select *h* to roughly based on dataset size and the hidden dimensions used by competing methods.
>
> > Q6: figure 5: can you move back from the latent space to the original data?
>
> **A6:** We thank the reviewer for this meaningful suggestion. While Figure 5 visualizes clusters in latent space that each prototype surrounded by its assigned queries, we can also compute the prototype's pattern in the input space by following method:
>
> 1. Collect all input sequences assigned to $P_k$ (i.e., those $X_i$ whose query $Q_i$ assigned to $P_k$).
> 2. Compute the average of these sequences to form the prototype’s representative pattern curve.
>
> Formally, it can be calculated as following equation:
> $$
> \text{Pattern}\_k = \frac{1}{|\{i: Q\_i \in P\_k\}|}\sum_{Q\_i \in P\_k} X\_i,
> $$
> where ($X_i$) denotes the input sequence whose query $(Q_i)$ assigned to prototype \($P_k$\).
>
> We then visualize the traffic speed patterns of prototypes and observe several distinct patterns on METR-LA dataset. Since NeurIPS does not allow image updates in rebuttal phase, we would like to describe some of the phenomena we discovered:
>
> - **Prototype 7, 12, 14**: Rapidly decreasing curves $\Rightarrow$ sudden‐drop events (i.e. traffic accident).
> - **Prototype 0, 2, 3, 4, 9, 13, 16, 18**: Flat curves at different levels $\Rightarrow$ stable traffic conditions with small fluctuation.
> - **Prototype 10, 15, 17**: Gradually decreasing curves $\Rightarrow$ morning or evening rush hour.
> - **Prototype 1, 5, 6,**: Increasing curves $\Rightarrow$ after morning or evening rush hour.
>
> Moreover, these representative patterns also **align well with the visualization of Figure 4**, further demonstrating that these prototypes capture meaningful spatio-temporal patterns. We commit to including this meaningful visualization and analysis in the final version to strengthen interpretability.
>
> **Reference**:
>
> [1] Dl-traff: Survey and benchmark of deep learning models for urban traffic prediction, CIKM, 2021.
>
> [2] Pre-training enhanced spatial-temporal graph neural network for multivariate time series forecasting, SIGKDD, 2022.
>
> [3] Z-GCNETs: Time zigzags at graph convolutional networks for time series forecasting, ICML, 2021
>
> [4] Diffusion convolutional recurrent neural network: Data-driven traffic forecasting, ICLR, 2018.
>
> [5] Graph WaveNet for Deep Spatial-Temporal Graph Modeling, IJCAI, 2019.
>
> [6] Adaptive graph convolutional recurrent network for traffic forecasting, NeurIPS, 2020.
>
> [7] Connecting the dots: Multivariate time series forecasting with graph neural networks, SIGKDD, 2020.
>
> [8] Discrete Graph Structure Learning for Forecasting Multiple Time Series, ICLR, 2021.
>
> [9] A neural attention model for urban air quality inference: Learning the weights of monitoring stations, AAAI, 2018.
>
> [10] Weather forecasting using ensemble of spatial-temporal attention network and multi-layer perceptron. *Asia-Pacific Journal of Atmospheric Sciences*, *57*(3), 2020.

---

> > ### Comment · Area_Chair_cGpj · 2025-08-04
> >
> > Dear Reviewer,
> >
> > Please respond to the rebuttal. Thanks.
> >
> > AC.

---

> > ### Comment · Reviewer_hTWY · 2025-08-04
> >
> > Dear Authors
> >
> > Thanks for your detailed rebuttal that addresses my major concerns. I will raise my score to 5.

---

> ### Author Response · Authors · 2025-08-05
>
> We are sincerely grateful for your constructive feedback throughout this process and for taking the time to reconsider your evaluation. Thank you very much for your support of our work.

---

> > ### Author Response · Authors · 2025-08-05
> > **Kind Reminder**
> >
> > **Kind Reminder: Thank you again for endorsing our work for acceptance (raising score to 5). We noticed that the final rating has not yet been submitted, so please don’t forget to submit it when it’s convenient for you.** 😊

---

### Author Response · Authors · 2025-08-08
**Summary of Reviews**

Dear Area Chair,

Thank you so much for your great efforts. As the rebuttal phase draws to a close, for your convenience, we would like to summarize the reviews and rebuttals as follows.

The majority of reviewers have acknowledged that our rebuttal well addressed their concerns and provided positive ratings **(5, 5, 4)** after the rebuttal period.

ST-SSDL introduces a novel and valuable perspective by formally addressing the previous overlooked problem of **dynamic deviation** between current observations and historical anchors in spatio-temporal forecasting. It quantifies deviation by two novel self-supervised losses that enforce a "relative distance consistency" in the latent space.

The state-of-the-art results on six benchmarks, combined with its ability to enhance diverse model architectures, demonstrate the effectiveness and robustness of our method.

Accordingly, we consider ST-SSDL worthy of publication at NeurIPS and believe it will make a valuable contribution to the spatio-temporal forecasting community.

Best Regards,

Authors of ST-SSDL

---

### Note · Authors · 2025-08-13

Dear Area Chair,

Thank you so much for your great efforts. We also sincerely thank all reviewers for their time and feedback. For your convenience, we would like to summarize the reviews and rebuttals as follows.

The majority of reviewers have acknowledged that our rebuttal well addressed their concerns and provided **positive ratings (5, 5, 4)** after the rebuttal period. We briefly summarize the key discussion points:

- For **Reviewer hTWY (4→5)**, we provided a detailed explanation of our core motivation and the idea of **"relative distance consistency"**. We further strengthened the paper with **new statistical significance tests** and a **new interpretability analysis**. The reviewer confirmed the major concerns were addressed and raised the score.
- For **Reviewer DAZY (5)**, we addressed questions on **robustness** and **generalizability** by clarifying how our method can handle outliers and irregular patterns. We also provided the new interpretability analysis, which the reviewer noted addressed the main concerns.
- For **Reviewer TcCg (3→4)**, we addressed the main concerns by conducting new experiments against **two recent SOTA baselines** (HimNet '24, MegaCRN '23) to confirm our performance. We also clarified our key novelties, including the fundamental differences from VQ-VAE and the proposed new concept **"dynamic deviation"**. The reviewer also raised the score.
- For **Reviewer CVvy (3)**, we also worked diligently to address the concerns. We clarified that our contribution lies in the **self-supervised modeling of deviation**, rather than the use of prototypes. Besides, we provided new experiments to demonstrate that SSDL can **significantly boost performance** for other backbones like MLP and Transformer.

ST-SSDL introduces a novel and valuable perspective by formally addressing the previous overlooked problem of **dynamic deviation** between current observations and historical anchors in spatio-temporal forecasting. It quantifies deviation by two novel self-supervised losses that enforce a "relative distance consistency" in the latent space. The state-of-the-art results on six benchmarks, combined with its ability to enhance diverse model architectures, demonstrate the effectiveness and robustness of our method.

Accordingly, we consider ST-SSDL worthy of publication at NeurIPS and believe it will make a valuable contribution to the spatio-temporal forecasting community.

Best Regards,

Authors of ST-SSDL

---

### Decision · Program_Chairs · 2025-09-17

**Decision:**

Accept (poster)

**Comment:**

This paper proposes ST-SSDL, a spatio-temporal forecasting framework that models dynamic deviations between current observations and historical anchors using prototype-based self-supervised learning. The method introduces novel deviation and contrastive losses to enforce relative distance consistency in latent space, achieving strong and consistent improvements across six traffic benchmarks with thorough ablations, interpretability analyses, and new baselines added during rebuttal. Reviewer scores (5, 5, 4, 3) reflected both strong enthusiasm for the contribution and lingering concerns about incremental novelty and evaluation scope. While one reviewer remained unconvinced about conceptual distinctiveness, others were persuaded by the rebuttal and emphasized the value of the deviation modeling perspective and the empirical rigor. On balance, this work represents a solid technical contribution with meaningful empirical impact.